

# Impact of a medicane on the oceanic surface layer from a coupled, kilometre-scale simulation

Marie-Noëlle Bouin[1,2] and Cindy Lebeaupin Brossier[1]

[1]CNRM UMR 3589, Université de Toulouse, Météo-France, CNRS, Toulouse, France.
[2]LOPS UMR 6523, Ifremer, CNRS, UBO, IRD, IUEM, Plouzané, France

**Correspondence:** M.-N. Bouin (marie-noelle.bouin@meteo.fr)

**Abstract.** A kilometre-scale coupled ocean–atmosphere numerical simulation is used to study the impact of the 7 November 2014 medicane on the oceanic upper layer. The processes at play are elucidated through analyses of the tendency terms for temperature and salinity in the oceanic mixed layer. Whereas comparable by its maximum wind speed to a Category 1 tropical cyclone, the medicane results in a substantially weaker cooling. As in weak to moderate tropical cyclones, the dominant

contribution to the surface cooling is the surface heat fluxes, with secondary effects from the turbulent mixing and lateral advection. Upper-layer salinity decreases due to heavy precipitation that overcompensates the salinizing effect of evaporation and turbulent mixing. The upper-layer evolution is marked by several features believed to be typical of Mediterranean cyclones. First, strong, convective rain occurring at the beginning of the event build a marked salinity barrier layer. As a consequence, the action of surface forcing is favoured and the turbulent mixing dampened, with a net increase of the surface cooling as result.

Second, due to colder surface temperature and weaker stratification, a cyclonic eddy is marked by a weaker cooling, oppositely to what is usually observed in tropical cyclones. Third, the strong dynamics of the Sicily Strait enhances the role of the lateral advection in the cooling and warming processes of the mixed layer.

## 1 Introduction

Tropical cyclones (TCs) have been known for long to result in a strong cooling of the oceanic upper layer. In situ measurements

from ship (Leipper, 1967) then using buoy arrays (Cione et al., 2000; D'Asaro, 2003) and more recently satellite observations (e.g. Lin et al., 2005; Chiang et al., 2011) showed that they generate a cold wake with maximum amplitude in their right-rear quadrant (in the northern hemisphere), which can lower the sea surface temperature (SST) of 5 to 6 °C. This surface cooling can have a strong feedback effect on the cyclone development and intensity, as SST controls for a large part the surface enthalpy flux (e.g. Bender et al., 1993; Schade and Emanuel, 1999). This typical ocean response (and feedbacks) has motivated many

studies documenting the processes at play and the factors controlling its amplitude.

The oceanic processes at the origin of the cooling have been investigated in case studies involving simulations (Price, 1981; Morey et al., 2006; Huang et al., 2009; Chen et al., 2010) or based on observations (Sanford et al., 1987; D'Asaro, 2003; D'Asaro et al., 2007; Vincent et al., 2012a). Turbulence injected into the oceanic mixed layer (OML) by the surface wind stress results in a very strong vertical shear at the base of the OML with respect to the underlying thermocline. For cyclones with a





translation speed above ∼1-2 m s$^{-1}$, near-inertial currents act at reinforcing this shear and establish a vertical oscillation with period slightly below the inertial period and amplitude of 20 to 40 m (Greatbatch, 1983; Shay, 2010). The first upwelling of this series of alternate vertical motions contributes to cool the upper-layer by bringing colder water from below the thermocline into the OML. The resulting change of the mixed-layer depth (MLD) can further alter the OML budget and its response to the atmospheric forcing. This turbulent mixing usually accounts for most of the cooling effect in strong cyclones (category 2 or

above in the Saffir–Simpson scale). A global study based on the simulation of more than 3000 TCs evaluated its contribution to 56 % of the SST cooling on average within 200 km of the TC track. This contribution varies depending of the TC intensity, from ∼ 30 % of the total cooling for the weakest cyclones to more than 90 % for the most intense ones (Vincent et al., 2012a). Surface heat fluxes also act at cooling the upper-ocean layer, mainly through the latent heat flux. They account for the remaining part of the surface cooling, up to 70 % for the weakest TCs but less then 10 % for the TCs of intensity 4 or above.

Lateral advection also contributes as a secondary process, accounting for nearly 10 % of the total cooling for the strongest TCs, and a larger proportion for weaker ones. Especially, it may reinforce the asymmetry of the cold wake originating from the shear-driven turbulence at the base of the OML (e.g. Price, 1981; Huang et al., 2009; Vincent et al., 2012a).

Cyclone intensity and translation speed are key criteria controlling the cooling magnitude and its spatial extent. More intense cyclones generate stronger cooling, and slowly moving cyclones produce stronger cooling for a given intensity (Lloyd and Vec-

chi, 2011; Mei and Pasquero, 2013). Slowly moving cyclones also produce cold wakes of larger extent, and more symmetrical because the upwelling has more time to settle (Vincent et al., 2012a).

The amplitude and spatial extent of the cold wake also depend on the ocean preconditioning, with a strong influence of the pre-storm SST, upper-layer stratification, and MLD. The amplitude of the cooling has been shown to vary of one order of magnitude depending on the ocean state prior to the cyclone passage (Vincent et al., 2012b). In strong TCs, shallower MLD

or stronger stratification favour more intense cooling as the turbulent mixing at the base of the OML more efficiently brings colder water upwards. Colder SSTs, or lower ocean heat content (generally defined as the temperature integral from the surface down to the 26 °C isotherm depth, (Leipper and Volgenau, 1972)) generate weaker cooling and limit the cyclone intensity.

Upper-layer oceanic features like fronts or eddies also have the potential to modulate the upper-ocean response to a TC. Statistical studies based on 30 to 3000 cases showed that both anticyclonic and cyclonic eddies can influence the TC development

through the upper ocean feedback (Lin et al., 2008; Jullien et al., 2014). Anticyclonic eddies deepen the OML and limit the SST cooling by insulating the warm surface layer against storm-induced upwelling and mixing. Conversely, cyclonic eddies are associated with shallower OML and lead to stronger surface cooling.

The surface cooling can also be altered by salinity stratification. Salinity-induced barrier layers (BLs, Godfrey and Lindstrom (1989)) form in the presence of less saline water at the surface, due to rain freshening, run-off or horizontal advection. They

uncouple the OML, driven by salinity, from the isothermal surface layer and form an intermediate barrier layer (at least 5 m deep) more saline than the OML and warmer than the water below the thermocline. A statistical study (Yan et al., 2017) showed that the effect of the BL strongly depends on the intensity (or stage) of the TC. For a TC of weak intensity, where most of the cooling arise from the surface heat fluxes, the BL acts at shoaling the ML and isolating it from the colder waters below and thus makes the surface heat extraction more efficient. For a TC of stronger intensity, with most of the cooling due to shear





instability and mixing at the base of the ML, the presence of the BL mitigates the efficiency of the mixing.

The upper-ocean salinity changes in the wake of a TC have motivated less studies than the surface cooling, because of their indirect effect on the cyclone intensity. However, they can modulate the subsurface stratification and enhance or weaken the mixing or heat extraction. Sea surface salinity (SSS) changes result from the competing effects of precipitation and vertical shear and mixing near the base of the OML bringing saltier water up to the surface. Rain rates in TCs are maximum 35 to 50 km

away from the storm centre and their magnitude ranges 3 to 12 mm h$^{-1}$ on average depending on the cyclone intensity (Lonfat et al., 2004). In a study of more than 800 TCs based on satellite observations and reanalysis of the upper-ocean characteristics, Jourdain et al. (2013) showed that, despite strong precipitation and neglecting the effect of evaporation, the wake of TCs is marked by salinization. Strong mixing brings saltier water from below the OML, overcompensating the freshening effect of rain. Note that, in this study, the SST cooling is attributed entirely to the turbulent mixing. A recent, more realistic study using

in situ observations brought contrast to that, showing that, at least in a 150 km radius around the cyclone centre, the freshening effect of strong precipitation dominates the saltening due to mixing and evaporation (Steffen and Bourassa, 2018).

The cooling of the oceanic upper layer by extratropical storms or hybrid cyclones like medicanes has motivated fewer studies than for TCs, probably because their magnitude is much smaller and there is supposedly no or a very small feedback on the atmosphere. Indeed, coupled simulations of midlatitude storms in the North Atlantic showed that, in contrast with TCs where

the cooling can reach several degrees, the mean effect is below 1° C (Ren et al., 2004; Yao et al., 2008). A recent statistical study on North Pacific storms, based on satellite observations and reanalysis over 20 years, gave also a mean SST cooling one order of magnitude smaller than what is obtained for TCs (Kobashi et al., 2019). Surface heat fluxes and turbulent mixing contributed equally to the cooling. Finally, a study of a strong storm in the Gulf of Lion, in May 2005, using a coupled ocean–atmosphere–waves simulation obtained a surface cooling of 2 °C over a large area (Renault et al., 2012). During this

storm, the major contributor to the OML cooling was the vertical mixing enhanced by the strong surface stress (68 %), with secondary contributions from the latent and sensible heat fluxes (15 and 7.5 % of the cooling). These contributions are similar to those obtained in TCs of Category 2 or above.

In the present study, we use of a coupled ocean–atmosphere kilometre-scale simulation to investigate the impact of the short but intense medicane Qendresa (7 November 2014) on the OML. Comparing the atmospheric processes of medicanes with those of

TCs is a tempting but challenging undertaking. Indeed, no systematic statistical study provides, to the best of our knowledge, a comprehensive assessment of the processes at play according to the cyclone intensity, especially for the Categories 1 and 2 that could be compared with medicanes. Conversely, as detailed here above, the impact of TCs on the upper ocean and the corresponding mechanisms were the subject of systematic studies scanning the different TC categories (e.g. Vincent et al., 2012a; Jullien et al., 2014). We thus take advantage of this knowledge to contrast the coupling mechanisms and oceanic upper-layer

processes in a medicane with those of TCs.

A first study of the lifecycle and atmospheric processes of this medicane, including the assessment of the impact of the SST cooling on the atmosphere, showed that the surface cooling is at least one order of magnitude lower than in typical TCs (Bouin and Lebeaupin Brossier, 2020). The present study aims at investigate i) how the surface cooling / salinity changes obtained in this case compare with changes observed in TCs or midlatitude storms; ii) whether the OML processes are similar to those of



TCs; iii) whether the characteristics of the changes obtained here are modulated by the atmospheric forcing or by the Central
Mediterranean oceanic conditions prior to the medicane. Section 2 summarizes the case study and presents the oceanic condi-
tions before the event, and the numerical tools used in this work. The results, with the evolution of the ocean and an analysis
of the role of the atmospheric forcings and of the mechanisms at play, are given in Section 3. Section 4 presents the role of the
oceanic conditions on the cooling and salinity changes in different areas. These results are discussed and conclusions are given

Section 5.

## 2   Case study and simulations

We present here a summary of the November 2014 medicane, the oceanic conditions that pre exist in Central Mediterranean,
and the ocean–atmosphere simulating configuration used in this study.

### 2.1   The November 2014 medicane

The November 2014 medicane has been extensively described in several case studies (Carrió et al., 2017; Pytharoulis, 2018;
Cioni et al., 2018; Bouin and Lebeaupin Brossier, 2020), only a short summary of its lifecycle is given here.

#### 2.1.1   Synoptic situation

The medicane formed north of Lampedusa in the morning of 7 November, from the conjunction of a baroclinic disturbance
at low level and of upper-level instability. Strong convection developed in the morning with heavy precipitation (more than
150 mm locally) in the Sicily area. The low-level system rapidly deepened, with a sudden drop of sea-level pressure of 8 hPa
in 6 hours, and evolved into the quasi-circular structure of a tropical cyclone with spiral rain bands and a cloudless eye-like
centre. The maximum intensity was reached around 12 UTC on 7 November north of Lampedusa, with sustained 10 m wind
speeds above 34 m s$^{-1}$. Strong winds persisted during its transit towards the Sicilian south coasts, with a landfall at Malta
around 17 UTC. It reached Sicily in the evening of 7 November and continued its decay during the following night on the
Ionian Sea close to the Sicily coasts, until 12 UTC on 8 November. Three distinct phases of the event are described in Bouin
and Lebeaupin Brossier (2020): the development phase with strong convection and a maximum of precipitation until 11 UTC
on 7 November, where air–sea exchanges of heat and momentum are maximum; the mature phase until 18 UTC the same day,
with surface wind decaying but strong surface heat fluxes and heavy rain persisting; and the decay phase until 12 UTC on 8
November, where surface exchanges rapidly decrease. The same study shows that the surface heat transfer throughout the event
are mainly controlled by the surface wind and the SST.

The trajectory of the simulated event is given Fig. 1, as well as the bathymetry on the domain of the study (see Cioni et al.
(2018) and Fig. 2 of Bouin and Lebeaupin Brossier (2020) to compare to the best track obtained from satellite data analysis).





### 2.1.2 Oceanic conditions

The simulation after a spin up of four hours is used here to describe the oceanic conditions prevailing at the beginning of the
event.

The simulated medicane spent most of its lifetime in the Sicily Strait area, which is significantly shallower than the surrounding basins with a mean bathymetry close to 500 m and large areas shallower than 100 m, for instance in the Gulf of Gabès. As a transition between the Western and Eastern Mediterranean, the region is characterized by large-scale gradients (Drago et al., 2010): a north-south thermal gradient and a west-east salinity gradient between Atlantic Water flowing eastwards along the
Tunisian coasts (AW, 0 to 100 m depth, T = 15–17 °C, S = 37.2–37.8 with a salinity minimum around 50 m) and Ionian Water (from 50 to 100 m depth, T = 15–16.5 °C, S = 37.8–38.4). These two water masses cap the Levantine Intermediate Water (LIW, with a core depth at 300 m, T = 13.75–13.92 °C, S = 38.73–38.78) that originates from the thermohaline circulation in the Eastern Basin and flows westwards. These gradients are well represented in the surface initial conditions used in this study (Fig. 2 and 3b). North of the Sicily Strait, the SST is below 21 °C, while it reaches 24 °C close to the Libyan coasts.
This marked SST contrast has been shown to largely control the surface heat fluxes during the most intense part of the event, along with the surface wind speed (Bouin and Lebeaupin Brossier, 2020). In the Tyrrhenian Sea and the Sicily Strait, the SSS is below 38, with a strong impact of the AW flowing eastwards along the Tunisian coasts (the AW flow can also be seen on the SST map).

When comparing the model initial SSTs (at 01 UTC on 7 November) with those provided by a satellite analysis at 00 UTC on 7
November, the model SSTs are biased low of $-0.63 \pm 0.49$ °C, but the general patterns are well represented with a correlation coefficient ($r^2$) of 0.86.

On average for this area, the climatology from d'Ortenzio et al. (2005) gives MLDs between 15–30 m and 40–50 m in October and November, respectively. This is consistent with the MLD at the beginning of the simulation (Fig. 3d), between 10 and 40 m in the Tyrrhenian and Ionian Seas, and between 40 and 50 m in most of the Sicily Strait and North of Libya. Note that the
MLD used in this study is defined as the depth where the density is equal to $\rho_0 + \Delta\rho_c$, with $\rho_0 = \rho_{-10}$ the density at 10 m for each grid point and $\Delta\rho_c$ the density criterion of 0.10 kg m$^{-3}$ here.

The surface circulation (Fig. 3c) shows a large mesoscale variability with coastal jets, gyres, meanders and filaments (Poulain and Zambianchi, 2007; Ciappa, 2009). These structures are generated by perturbations induced by tidal, inertial, gravity, surface and continental shelf waves and non-linear interactions with bathymetry and/or by synoptic scale atmospheric forcing and local
wind regimes (Hamad et al., 2006; Omrani et al., 2016). Several eddies are supposed to be present regularly like the cyclonic eddy south west of Sicily (Adventure Bank Vortex, ABV hereafter, (Sorgente et al., 2011)) and the anticyclonic Sorgente Gyre north of the Libyan coasts, also visible here. The ABV is marked by SSTs significantly colder than the surrounding area (20.0 $\pm$ 0.4 °C with respect to 21.5 $\pm$ 0.5 °C). The OML in the eddy is also shallower than in the rest of the Sicily Strait with 20 $\pm$ 4 m with respect to 36 $\pm$ 11 m elsewhere.

The stratification index (SI) used in this study is computed as in the work of Estournel et al. (2016) as the amount of buoyancy in kg m$^{-2}$ to be extracted to mix the water column from the surface to level $z$ (here $-100$ m, chosen to represent the stratification




of the surface layer) and achieve a homogeneous density $\rho$:

$$SI_{-100} = \int_{-100}^{0} (\rho(-100) - \rho(z))dz \tag{1}$$

At the beginning of the simulation (Fig. 3a), the stratification is pronounced in the Sicily Strait with an influence of the AW (SI
= $125 \pm 14$ kg m$^{-2}$) and weaker in the Ionian Sea ($92 \pm 11$ kg m$^{-2}$). Minimum values are visible on the Tunisian continental
shelf (around the Gulf of Gabès, with depth less than 30 m, see Fig. 1). Continental shelf waters are known to be significantly
more sensitive to atmospheric conditions (Béjaoui et al., 2019) with wind-induced mixing that quickly homogenizes the water
column, and salinity reacting instantaneously to evaporation, runoff or heavy rain (Ismail et al., 2017).

## 2.2 Numerical settings and tools

The ocean–atmosphere coupled numerical simulation of the event was performed using the state-of-the-art atmospheric model
Meso-NH (Lac et al., 2018) and oceanic model NEMO (Madec and the NEMO team, 2015).

### 2.2.1 Atmospheric model

The non-hydrostatic French research model Meso-NH version 5.3.0 is used in the present study with numerical and physical
packages as described in Bouin and Lebeaupin Brossier (2020). The radiative transfer is computed by solving long-wave and
short-wave radiative transfers separately using the European Centre for Medium-range Weather Forecasts (ECMWF) opera-
tional radiation code (Morcrette, 1991). The sea-surface fluxes are computed within the SURFEX module (Surface External-
isée, Masson et al. (2013)) using the iterative bulk parameterization ECUME (Belamari, 2005; Belamari and Pirani, 2007)
linking the surface turbulent fluxes to the meteorological gradients and the SST through the appropriate transfer coefficients.
The Meso-NH model shares its physical representation of parameters, including the surface fluxes parameterization, with the
French operational model AROME used for the Météo-France weather prediction with an horizontal resolution of 1.3 km (Se-
ity et al., 2011). In the present study, a first atmosphere-only simulation at the horizontal resolution of 4 km has been run on a
larger domain of 3200 km × 2300 km (not shown here) to provide initial and boundary conditions for the simulation at 1.33
km on a smaller domain of 900 × 1280 km (Fig. 1). This run at coarser resolution started at 18 UTC on 6 November and lasted
42 h until 12 UTC on 8 November. Its initial and boundary conditions come from the ECMWF operational analyses every 6
h. The coupled simulation on the inner domain that is used here to investigate the oceanic evolution starts at 00 UTC on 7
November and lasts until 12 UTC on 8 November.

### 2.2.2 Oceanic model

The ocean model used is NEMO (version 3_6) with physical parameterizations as follows. The total variance dissipation
scheme is used for tracer advection in order to conserve energy and enstrophy (Barnier et al., 2006). The vertical diffusion fol-
lows the standard turbulent kinetic energy formulation of NEMO (Blanke and Delecluse, 1993). In case of unstable conditions,
a higher diffusivity coefficient of 10 m$^2$ s$^{-1}$ is applied (Lazar et al., 1999). The sea surface height is a prognostic variable





solved thanks to the filtered free-surface scheme of Roullet and Madec (2000). A no-slip lateral boundary condition is applied and the bottom friction is parameterized by a quadratic function with a coefficient depending on the 2D mean tidal energy (Lyard et al., 2006; Beuvier et al., 2012). The diffusion is applied along iso-neutral surfaces for the tracers using a laplacian

operator with the horizontal eddy diffusivity value $\nu_h$ of 30 m$^2$s$^{-1}$. For the dynamics, a bi-Laplacian operator is used with the horizontal viscosity coefficient $\eta_h$ of $-1.10^9$ m$^4$s$^{-1}$.

The configuration used here is sub-regional and eddy-resolving, with a 1/36° horizontal resolution over an ORCA-grid (from 2 to 2.6 km resolution) named SICIL36 (tripolar grid with variable resolution Madec and Imbard, 1996), that was extracted from the MED36 configuration domain (Arsouze et al., 2013) and shares the same physical parameterizations with its "sister"

configuration WMED36 (Lebeaupin Brossier et al., 2014; Rainaud et al., 2017). It uses 50 stretched $z$-levels in the vertical, with level thickness ranging from 1 m near the surface to 400 m at the sea bottom (i.e. around 4000 m depth) and a partial step representation of the bottom topography (Barnier et al., 2006). It has 4 open boundaries corresponding to those of the domain shown in Fig. 1, and its time step is 300 s. The open boundary conditions come from the global 1/12° resolution PSY2V4R4 daily analyses from Mercator Océan International (Lellouche et al., 2013) also used as initial conditions at 00 UTC on 7

November.

### 2.2.3  Configuration of the coupled simulation

The coupled simulation between the Meso-NH and NEMO-SICIL36 models relies on the SURFEX-OASIS coupling interface developed by Voldoire et al. (2017). The two components of the momentum flux, the solar and non-solar heat fluxes and the freshwater flux are transmitted from the atmospheric model to the oceanic model every 15 min. The oceanic model exports

the SST and two components of the surface current into the atmospheric model at the same time step. Examining the time evolution of the oceanic parameters in the first hours of the simulation shows that there is a spin-up effect of the model with a strong adjustment of the ocean upper layer in the first hour. In the present study, we consider the evolution of the oceanic parameters after 01 UTC on 7 November.

## 3  Atmospheric forcings, evolution of the ocean, atmospheric forcings and processes

### 3.1  Atmospheric forcings

Here are described the characteristics and time evolution of the atmospheric forcings as seen by the ocean model. Among the forcing parameters, the solar and non-solar heat fluxes (heat fluxes hereafter) include the turbulent fluxes (latent and sensible heat fluxes). The water flux corresponds to precipitation minus evaporation, a positive water flux corresponds therefore to freshwater input into the ocean.

Because of the small size and short duration of the event, the methodology commonly used in TC studies (i.e. averaging values on circles of 500 km around the cyclone centre or contrasting mean SST values 3 days after and before the cyclone) is not appropriate here. Instead, we define and use in the following a strong fluxes area (SFA) as the area where both the accumulated





surface heat fluxes and the accumulated wind energy flux injected into the ocean at 07 UTC on 8 November are above their

80 % quantiles (this time corresponds to the end of the period of intense forcing, see Fig. 4). The wind energy flux (WEF) is

defined here as the scalar product of the wind stress by the surface currents (see Giordani et al. (2013) for a similar use). The

threshold values are $-42.8$ MJ m$^{-2}$ and $7.59 \ 10^3$ kg s$^{-2}$, respectively. Only the model grid points with bathymetry deeper

than 100 m and located in the Sicily Strait are kept and this results in 6574 points (see Fig. 6).

Over the SFA, the upward surface heat flux intensifies until 09 UTC on 7 November (median value $-723$ W m$^{-2}$, 5 % and

95 % quantiles $-117$ and $-985$ W m$^{-2}$, and decays slowly after that (Fig. 4a). Its median value stays close to $-400$ W m$^{-2}$

until 06 UTC on 8 November, but with less spreading and no values below $-800$ W m$^{-2}$. After 06 UTC on 8 November, the

turbulent fluxes extracting heat are close to zero and the solar flux is strong again, making the neat heat flux increasing rapidly

to positive values.

Conversely, the WEF grows continuously until 11-12 UTC (time of the maximum intensity of the medicane; Fig. 4), with a

median value of 0.25 kg s$^{-3}$, a 95 % quantile of 0.74 kg s$^{-3}$). After that time it drops more quickly than the heat flux and stays

around 0.1 kg s$^{-3}$ from 15 UTC on 7 November.

The time evolution of the water flux in the SFA is largely dominated by precipitation until 21 UTC on 7 November (Fig.

4c). Conversely to the heat and momentum fluxes where the distributions are almost symmetrical (the mean and median values

almost coincide), the water flux distribution is strongly asymmetric towards the highest values on 7 November. The largest water

flux in mean and median values occurs at 10 UTC on 7 November (4.6 mm h$^{-1}$ and 2.0 mm h$^{-1}$ respectively). Conversely, the

highest water flux (95 % quantile) occurs before 06 UTC on 7 November with a 30 minutes averaged value of 24.8 mm h$^{-1}$.

In the SFA, the time evolution of the heat fluxes, WEF and water flux is in contrast with what is generally observed in TCs.

The maximum sustained wind speed of Qendresa makes it comparable to a Category 1 TC on the Saffir–Simpson scale. The

values of the heat fluxes exceed values usually obtained for Category 1 TCs. They peak earlier than the WEF and stay above

400 W m$^{-2}$ for more than 24 h. The water flux is strong, always positive, with values comparable to Category 2 TCs. Heavy

rainfall occurring in the early hours of 7 November result in a freshwater input around 5 mm h$^{-1}$ until 10 UTC.

### 3.2  Sea surface cooling and salinity change

In the simulation, the studied domain is affected by an overall weak surface cooling of $-0.18 \pm 0.21$ °C over the first 24 hours

(Fig. 5b). A comparison of the SST change with satellite observations from the Group for High-Resolution SST (GHRSST)

L4 products (Piolle et al., 2010) shows an overestimation of the cooling by the model (mean bias of $-0.26 \pm 0.22$ °C, see Fig.

5). The largest surface cooling is obtained southeast of the Sicily Strait, between the north of Tunisia and the south of Sicily,

where the medicane spent several hours with low translation speed and strong air–sea surface exchanges in the morning of the

7 November. In the SFA, the mean SST cooling at 00 UTC on 8 November is $-0.56 \pm 0.24$ °C (maximum $-1.37$ °C, minimum

0.20 °C) and the mean cooling of the OML ($\langle\theta\rangle$ hereafter) is $-0.54 \pm 0.23$ °C (Table 2). The evolution of the salinity within the

OML ($\langle S \rangle$) shows more contrast, with a very weak median change of $-0.010$ over the whole domain, a 5 % quantile positive

change of 0.053 and a 95 % quantile negative change (freshening) of $-0.110$. At the surface, the salinity is almost constant on

average ($-8.3 \ 10^{-3}$) with a strong spatial variability (Fig. 6). The overall effect is therefore a freshening, with some areas of





salinification northwest of Sicily and along the coasts of Tunisia, for instance in the shallow area of the Gulf of Gabès where the evaporation is the most efficient. The OML freshening dominates in the SFA, with a mean value of $-0.07$, a maximum positive change (5 % quantile) of 0.03 and a maximum negative change (95 % quantile) of $-0.20$. Within the SFA, the spatial

variability of the salinity change is also substantial.

The SST cooling obtained here is lower than for TCs of Category 1 where wakes 1 to 2 °C colder than the surrounding ocean are currently observed (e.g. Vincent et al., 2012a). The weak surface freshening we obtain is rather consistent with the work of Steffen and Bourassa (2018) based on Argo observations showing that within 200 km around the TC centre, freshening overcomes salinification in every basin. This freshening occurs early in the medicane life cycle, with two thirds of the final

value reached at 12 UTC on 7 November, due to the strong precipitation in the morning of 7 November.

### 3.3 Oceanic processes

This part describes the simulated OML processes obtained throughout the event, with reference to the processes commonly observed in TCs. As seen in Section 2.1.2, the SFA represents a strongly dynamic region with large horizontal gradients of temperature and salinity, strong currents and the presence of the ABV cyclonic eddy. The stratification index in the SFA is high

at the beginning of the simulation with values of $125 \pm 14 \, \text{kg m}^{-2}$ except in the ABV cold eddy ($88 \pm 14 \, \text{kg m}^{-2}$, Fig. 3, Table 1). The MLD is $34 \pm 11$ m around the eddy, $16 \pm 3$ m in the eddy itself. At 13 UTC on 7 November, the cyclonic circulation at 15 m within the eddy has reinforced due to strong wind stress, with a marked diverging component and maximum velocities between 0.7 and 0.8 m s$^{-1}$. As a consequence of this subsurface divergence, the sea surface height has decreased of 10 cm in the eddy centre and strong upward motions develop under the MLD, around 50 m depth. The strong shear resulting from

the horizontal currents close to the bottom of the OML generates violent mixing, which brings colder water from below the thermocline up. The upwelling is maximum at 15 UTC, with an ascent of the thermocline of 5 m, and a lowering of the SSH of 17 cm. Following this, vertical oscillations establish with period close to 19 h (the inertial period for this area is between 20.5 and 21 h). At 00 UTC on 8 November, the amplitude of the subsurface cyclonic circulation has returned to its initial value, but with a marked converging component. At that time, the vertical velocity under the OML is negative, and the SSH is still 10 cm

depressed. The OML is deeper at the eddy centre, and shallower around the outer edge of the eddy.

The processes obtained are similar to those observed in TCs. To assess the respective roles of the surface heat fluxes and of the turbulent mixing in the time evolution of the OML, we performed a budget analysis of the temperature and salinity in the OML, using the equation of the tracer tendency as in Vialard and Delecluse (1998). The tendency of a given tracer $X$ ($\theta$ or $S$)




within the OML is given by

$$
\begin{aligned}
h\frac{\partial\langle X\rangle}{\partial t} = \quad & \mathcal{F}^X && \textbf{[FOR]}\times h \\
& -h\langle\overrightarrow{U}.\overrightarrow{\nabla}X\rangle && \textbf{[h-ADV]}\times h \\
& -\mathrm{w}_{(z=-h)}(\langle X\rangle - X_{(z=-h)}) && \textbf{[v-ADV]}\times h \\
& +h\left(\frac{\partial}{\partial x}\left(A^h\frac{\partial\langle X\rangle}{\partial x}\right)+\frac{\partial}{\partial y}\left(A^h\frac{\partial\langle X\rangle}{\partial y}\right)+\frac{\partial}{\partial z}\left(A^v\frac{\partial\langle X\rangle}{\partial z}\right)\right) && \textbf{[DIF]}\times h \\
& -\left(\underbrace{\frac{\partial h}{\partial t}+\overrightarrow{U}.\overrightarrow{\nabla}h}_{=0}\right)(\langle X\rangle - X_{(z=-h)})+\overline{\mathrm{w'}X'}_{(z=-h)} && \textbf{[ENT]}\times h
\end{aligned}
\tag{2}
$$

where $h$ is the MLD, $\overrightarrow{U}$ is the horizontal velocity vector, and w the vertical velocity. $A^h$ and $A^v$ are the horizontal and vertical eddy diffusivity coefficients ($\mathrm{m^2\,s^{-1}}$), $\mathcal{F}^X$ the surface flux, and $\overline{\mathrm{w'}X'}$ the turbulent flux of tracer $X$. Finally, $\langle X\rangle = \frac{1}{h}\int_{-h}^{0}X\,dz$. From that, the different terms of the temperature and salinity tendencies can be deduced:

- The surface forcing term (FOR) corresponds to the net heat flux for the potential temperature $\langle\theta\rangle$ (shortwave, longwave radiative net fluxes plus latent and sensible heat fluxes) and to the water flux for the salinity.

- The horizontal advection h-ADV that can be decomposed into zonal (ADV-X) and meridional (ADV-Y) advections.

- The vertical advection across the OML v-ADV.

- The diffusion term (DIF), that can decomposed into lateral and vertical diffusions, this latter term corresponding to the turbulent mixing within the OML (TM hereafter).

- The residual term (ENT) is linked to the variations of the surface defined by $z=-h$ and to the turbulent mixing at the
base of the OML.

More details and additional examples of applications can be found in the work of Lebeaupin Brossier et al. (2013) or Lebeaupin Brossier et al. (2014).

These tendency terms are used in the following to quantify the processes controlling the evolution of the temperature and salinity within the OML.

**3.3.1 Temperature tendency**

The time evolution of tendency terms for $\langle\theta\rangle$ in the SFA is given Fig. 7, with the corresponding evolution of $\langle\theta\rangle$. The continuous decrease of $\langle\theta\rangle$ throughout the simulation is mainly due to the surface forcing FOR. The evolution of this FOR term mimics the one of the median/mean values of heat fluxes in the SFA (Fig. 4b), with a maximum at 09 UTC. The TM evolution is closer to the evolution of the median value of the WEF (Fig. 4a), with a maximum between 10 and 12 UTC on 7 November.
Other significant contributions to the thermal evolution of the OML are from the lateral advection terms ADV-X and ADV-Y, which contribute to alternatively cool and warm the OML with a time period driven by the quasi inertial oscillation. The





vertical advection, lateral diffusion and entrainment terms do not contribute significantly. At 10 UTC on 7 November, where the cooling tendency is maximum in the SFA, the relative contribution of FOR and TM are −0.53 °C per day and −0.22 °C per day respectively, representing 53 % and 22 % of the cooling. The accumulated contributions to the cooling of the OML at

12 UTC on 8 November represent 65 % for FOR, 14 % for TM and 18 % for the horizontal advection. Contrasting with what is observed in intense TCs, the surface forcing is key here, the turbulent mixing playing only a secondary role like the lateral advection. The time evolution of the distribution of the FOR and TM terms for temperature in the SFA (Fig. 8) shows that all the quantiles of FOR are higher than those of TM. During the development phase and the beginning of the mature phase of the medicane (between 06 and 12 UTC on 7 November), TM even contributes to warm the OML for 25 % of the SFA. This is the

consequence of the ascent of the MLD under the effect of a surface salinity change. This is discussed in more details Section 3.3.2. Due to the time lag between the peaks of the heat flux and of the WEF, FOR continues to significantly cool the OML until 09 UTC on 8 November.

The spatial distribution of the cooling due to FOR and TM within the SFA is also different. A snapshot at 10 UTC on 7 November when the effect of TM is maximum (Fig. 9) shows the values of the FOR and TM terms for temperature averaged

on radial bins around the medicane centre. The turbulent mixing effect is stronger than the surface forcing one within 20 km around the medicane centre, but it drops rapidly and becomes small after 200 km. Conversely, the effect of FOR stays above −0.2 °C within 500 km around the centre.

The lateral advection contributes to alternatively cool and warm the OML (Fig. 7a) in phase with the near-inertial oscillation. Throughout the event, both ADV-X and ADV-Y cool the OML by −0.12 °C, or 16 % of the total cooling. A compensating

warming effect of ADV-X corresponds to 0.06 °C.

These results are consistent with those of the statistical study of Vincent et al. (2012a). First, the relative part of the surface forcing is higher than its of the turbulent mixing, as expected for the weakest TCs (70 % for FOR for TCs of Category 1 or below). Second, turbulent mixing and surface fluxes contribute equally within 2 radii of maximum wind, but surface fluxes dominate outside of that and concern a wider area. A case study in the Gulf of Mexico also showed that the surface heat fluxes

are at the origin of a widespread, moderate cooling affecting the whole surface of the gulf, while vertical mixing results in stronger, more localized cooling within 200 km around the cyclone centre (Morey et al., 2006). Third, the lateral advection contribute significantly to the cooling (and weakly to the following warming). This latter effect is likely due to the strong dynamics and horizontal gradients of the Sicily Strait.

In the following, we document the time evolution of the salinity in the OML.

### 3.3.2 Salinity tendency.

The time evolution of the OML salinity tendency terms in the SFA is given Fig. 10, with the corresponding evolution of the OML mean salinity and of the precipitation. The overall effect is a significant freshening until 21 UTC on 7 November, as long as the precipitation rate is above 2 mm h$^{-1}$. This freshening is thus due to the FOR term, and is partly compensated by TM and, as secondary processes, by the horizontal advection and entrainment. This compensating effect drops rapidly after 12

UTC on 7 November with the decrease of the WEF (Fig. 4) whereas the surface forcing generates freshening until 21 UTC. In





addition, the large amount of precipitation in the early hours of the 7 November deepen the BL by shoaling the mixed layer. During the event, the depth of the thermocline (defined by the depth where $\theta$ is equal to the 10 m-temperature minus 0.3 °C) averaged over the SFA oscillates of a few meters (Fig. 10b) due to the near inertial waves. The freshening due to heavy rain in the morning of 7 November makes the MLD defined by a density criterion shallower and governed by salinity rather than

temperature. As the OML moves 2 m upwards between 04 and 07 UTC on 7 November, its mean temperature slightly increases under the apparent effect of turbulent mixing - this actually corresponds to the points with warming tendency due to TM in Fig. 8. The BL thickness increasing from 5.8 m at 04 UTC to 8.1 m at 12 UTC on 7 November insulates the OML from the colder water below the thermocline. It enhances the efficiency of the surface heat extraction, and reduces the effect of the turbulent mixing.


## 4   Role of preconditioning and oceanic evolution

To quantify the role of oceanic preconditioning and precipitation in the oceanic response to the medicane, we define different zones within the SFA. The EDDY zone is defined as the area of the ABV cyclonic eddy, using a SST of 20.5 °C as a threshold (1005 grid points). We also define a heavy rain zone (HR hereafter) as the area, in the Sicily Strait, where integrated water flux

at 00 UTC on 8 November reaches 90 mm of water. This HR zone includes 2383 grid points and its mean integrated water flux at 12 UTC on 8 November is 136 mm. Note that it is not entirely within the SFA (Fig. 6). Finally, a reference zone includes the grid points of the SFA that are not part of either the EDDY or the HR zone (REF hereafter, 4343 grid points). The objective here is to evaluate whether the characteristics of Central Mediterranean (bathymetry, dynamics) or heavy rain control the oceanic response to the event.

### 4.1   Effect of the cyclonic eddy


The time evolution of the SST and OML temperature, in the zones EDDY and REF, is shown Fig. 11a. The corresponding values at 00 UTC on 8 November are given Table 2. All along the simulation, the SST and $\langle\theta\rangle$ in both EDDY and REF evolve similarly with a continuous decrease. The cooling of the OML in EDDY is weaker than in REF ($-0.32 \pm 0.25$ °C versus $-0.54$ $\pm 0.23$ °C at 00 UTC on 8 November) as is the integrated surface heat flux (Fig. 11b). The weaker cooling in EDDY originates

from lower heat fluxes throughout the event and especially between 09 and 16 UTC on 7 November (see also Table 3 for the integrated values of the surface heat flux, WEF and water flux at 12 UTC on 8 November). Lower fluxes result from the difference of SST in the two zones (20.0 $\pm 0.4$ °C for EDDY versus 21.6 $\pm$ 0.5 °C for REF at the beginning of the simulation) but also from the difference of wind speed (median WEF at 10 UTC on 7 November 0.11 kg s$^{-3}$ for EDDY versus 0.20 kg s$^{-3}$ for REF). Other differences include a significantly shallower MLD in EDDY (16 m versus 36 m in REF) and a weaker 100 m

stratification at the beginning of the simulation with a SI of 88 $\pm$ 14 kg m$^{-2}$ versus 125 $\pm$ 12 kg m$^{-2}$ in REF. The tendency terms for $\langle\theta\rangle$ (Fig. 12 and Table 3) confirm that different surface forcing leads to different cooling in EDDY and REF. In EDDY, the time evolution of FOR mimics the heat fluxes (red curves in Fig. 12a), with almost no forcing between





10 and 14 UTC on 7 November. The contribution of TM is stronger than in REF, especially between 08 and 14 UTC on 7

November. Yet, the WEF is much weaker in EDDY than in REF, the integrated value at 12 UTC on 8 November represents one

third of it. The stronger mixing is EDDY is due to a weaker stratification and a much shallower −0.3 °C thermocline (around

24 m versus 37 m in REF). This is consistent with the results of Jullien et al. (2014) showing that in stronger TCs, surface

cooling is promoted by cyclonic eddies. Here, the cooling is weaker in EDDY than is REF, but the part of the cooling due to

mixing (which is dominant in intense TCs) is larger. The horizontal advection significantly contributes to alternatively cool and

warm the OML (total cooling of −0.20 °C by ADV-Y, total warming of 0.23 °C by ADV-X in EDDY). In REF, it cools the

OML of −0.12 (ADV-X) and −0.14 °C (ADV-Y) like in the SFA.

Within EDDY, the salinity does not change significantly throughout the simulation (Fig. 13a, Table 2). Except between 09 and

11 UTC on 7 November, the rainfall amount in REF is weak (44 % of the integrated values of the SFA). However, the integrated

water flux (dominated by precipitation) is even much weaker in EDDY (9.6 versus 27.7 mm at the end of the simulation; Fig.

13c, Table 3). The tendency terms confirm that the precipitation drives the salinity evolution similarly in both zones (Fig. 14a

and c), with a largely compensating turbulent mixing (bringing saltier water from under the OML). In both zones, the OML

deepens of a few meters at the beginning of the event due to the cooling effect (Fig. 13b).

Within the ABV eddy, colder SSTs and shallower MLD and thermocline result in less surface cooling. The turbulent mixing

intensifies due to the weaker stratification, but the surface forcing is reduced, resulting in less overall cooling. Lateral advection

contributes more to the cooling and warming of the OML than outside of the eddy. Low precipitation and moderate mixing

bringing more saline water upwards compensate each other and do not change notably the OML salinity.

### 4.2   Role of heavy precipitation

We compare here the time evolution of the temperature and salinity in the OML, and the corresponding tendency terms, in the

HR and REF zones. At the beginning of the simulation, the OML is shallower in HR than in REF ($29 \pm 10$ versus $36 \pm 11$ m)

with SI at 100 m around 125 kg m$^{-2}$ in both cases. The SST is also similar ($\sim 21.5 \pm 0.5$ °C) and the surface water is slightly

fresher in HR ($37.61 \pm 0.12$ versus $37.73 \pm 0.17$). The time evolution of $\langle\theta\rangle$ is rather similar, with respective coolings at 00

UTC on 8 November of −0.57 and −0.51 °C (Fig. 11a, Table 2). Yet, the integrated heat loss in HR is 31 % higher in REF

than in HR (Fig. 11b, Table 3). Thus, the surface forcing is much more efficient in cooling the OML in HR than in REF (Fig.

12b and c). Looking at the tendency terms shows that FOR is stronger in HR than in REF, but that TM is stronger in REF than

in HR (Table 3). These discrepancies in the cooling effects in HR and REF can be related to the impact of the precipitation on

the upper-ocean salinity and the resulting MLD variations. In HR, the heavy rain in the morning of the 7 November produces

a first drop of the SSS of −0.1 (Fig. 13a) and shoals the OML of a few meters (Fig. 13b). As a result, the shallower OML is

more sensitive to the atmospheric forcing and is insulated from the colder waters below the thermocline by a 13 m thick BL

(Fig. 14b). As was shown for TCs, the effect of the surface heat fluxes on the SST is enhanced while the turbulent mixing is

dampened (Yan et al., 2017). Indeed, TM starts to decrease at 10 UTC in HR while the WEF increases until 12 UTC on 7

November. The net result here is a stronger cooling, since the FOR term is dominant because of the weak intensity of the event.

After this first freshening and cooling of the OML due the heavy rain in the morning of the 7 November, lighter rain occurs





in the afternoon. Yet, because the thinned OML is very sensitive to the atmospheric forcing at that time, this leads to strong additional freshening (−0.5 in total) and shoaling of the OML (Fig. 13). The water flux controlling the OML depth explains the stronger cooling in HR, with a SST cooling of 0.8 °C at 21 UTC on 7 November (Fig. 11).

This results are consistent with the role of BLs (usually pre existent) in modulating the surface cooling due to TCs. In weak TCs where the cooling effect is mainly due to the surface heat fluxes, the presence of BLs makes the OML shallower and enhances the surface forcing. Its isolating effect from colder water below the thermocline does not impact much the cooling since turbulent mixing is a secondary mechanism. The novel result here is that the BL formation and deepening arise from heavy precipitation occurring at the beginning of the event. A realistic study of the net effect of precipitation versus evapora-

tion and mixing on the surface salinity in TC wakes showed that the BL thickness increases in every cyclonic basin (Steffen and Bourassa, 2018). It is not clear however whether this overall freshening happens early enough during the TC development to substantially impact the cooling. Here, because of the deep convective rainfall in the first phase of the event (11 mm h$^{-1}$ on average during the first 10 h), a BL rapidly forms and strengthen the cooling effect of surface fluxes.

**5    Discussion and and conclusion**

Comparing a medicane with low-intensity TCs from the viewpoint of the upper-layer oceanic processes proves to be relevant and reveals many similarities and a few differences. According to its maximum sustained wind, the medicane studied here is comparable to TCs of Category 1 on the Saffir–Simpson scale. Its duration is nevertheless much shorter than a typical TC with wind energy flux above 0.2 kg s$^{-3}$ for a few hours only, between 07 and 13 UTC on 7 November, and upward heat fluxes above

400 W m$^{-2}$ for 30 hours between the beginning of the event and 07 UTC on 8 November. The mean surface cooling on the SFA is therefore less than −0.6 °C, significantly lower than the typical cooling of weak TCs. The dominant cooling mechanism is the surface forcing (at the origin of a −0.45 °C cooling) while the turbulent mixing accounts for less than −0.1 °C. These two mechanisms do not act simultaneously: the surface forcing starts earlier, peaks at 09 UTC on 7 November at stays above −400 W m$^{-2}$ for more than 24 h in total; the turbulent mixing increases until 12 UTC on 7 November and drops rapidly in

the afternoon of the 7 November. This may explain that surface forcing has a much larger effect than turbulent mixing on the OML temperature. Yet, the cooling effect of the surface fluxes is at the upper threshold of the typical range given by Vincent et al. (2012a), while the cooling effect of the turbulent mixing is always weak, even at the peak of intensity of the event. This discrepancy can be explained by the effect of heavy rainfall in the first hours of the event, which create or deepen an existing BL. Indeed, comparing the cooling of two areas forced by similar wind stress and heat fluxes, but very different water fluxes

shows that the presence of the BL results in a 37 % increase of the effect of the surface forcing and 64 % decrease of the effect of the turbulent mixing. A diagram based on the simulated mean profiles and MLD values summarizes the upper-layer evolution throughout the event, and the impact of heavy rain (Fig. 15). Without strong freshwater input, temperature governs the MLD evolution and the intense surface heat fluxes throughout the event continuously cool the upper-layer (Fig. 15a). This deepens the OML of a few meters (Fig. 15c) and moderate rain, partly balanced by evaporation, slightly freshen the OML (Fig.





15b). Near the bottom of the OML, turbulent mixing and advection further decrease the temperature. Heavy precipitation, when present, mainly acts on the surface temperature during the second half of the event (Fig. 15a, see also Fig. 11) even though it occurs since the beginning of the event. This delayed effect is due to the positive feedback of a salinity-induced thinning of the OML, which settles after a few hours only. Heavy convective precipitation are usual during the development phase of medicanes, before the wind reaches its maximum speed, and the preconditioning effect of the oceanic upper layer we obtain

here is probably rather typical. Mediterranean cyclones in general come with heavy rain, which are susceptible to alter the response of the oceanic upper layer as is observed here. On the OML salinity, both the effect of the surface forcing and of the turbulent mixing are enhanced by the BL.

Lateral advection plays a secondary but significant role in cooling or warming the OML. In particular, strong upper-layer horizontal currents develop close to the cyclone centre in the ABV eddy. Colder water is shifted southwards, reinforcing the

cooling, and warmer water is brought towards the eddy centre by converging motions from the surrounding area with deeper OML. In addition, due to shallower MLD and weaker 100 m stratification in the eddy, the effect of the surface heat fluxes is lower while the effect of the turbulent mixing is strengthened. As a result, the cooling in the ABV eddy is dampened with respect to the surroundings. The peculiar dynamics of the Sicily Strait, with its strong currents and the regular presence of eddies is able to modulate the impact of cyclones on the ocean.

The present study is in strong contrast with the case study of a strong storm with comparable sustained wind speed, in North-western Mediterranean (Renault et al., 2012). The cooling obtained by these authors was between $-1.5$ and $-2$ °C over the large area of the Gulf of Lion. A precise comparison of the two events and of their impact on the oceanic upper layer is beyond the scope of the present study but several factors can partly explain the stronger cooling they obtained. First, the duration of the storm was longer than Qendresa, with strong wind stress and heat fluxes from the 4 May until the 6 May. The track of the

storm was also favourable to a stronger impact on the ocean, since it looped for three days over the Gulf of Lion. Second, the MLD in May in the Gulf of Lion is usually shallower than in November in the Sicily Strait, with values in the range 10-30 m (d'Ortenzio et al., 2005). More precisely, observations of the Lion buoy show that prior to the storm, the temperature at 10 m was $-0.3$ °C below the SST (taken at 2 m, see Houpert et al., 2016). No measurements were available at that time at depths between 10 and 200 m, but observations at the same date in 2012 show a $-0.6$ °C difference between the temperature at 15

m and the SST, and a $-1$ °C difference at 35 m. The upper-layer was then likely strongly stratified before the storm, and this favours the turbulent mixing in bringing cold water at the surface. As far as we can judge, atmospheric and oceanic conditions acted together to enhance the impact of the turbulent mixing and this resulted in a stronger cooling.

This study is based on a single case, and our results should be confirmed by simulating more events. A broader study involving a statistically significant number of Mediterranean cyclones is nevertheless still challenging. Intense cyclones like the present

case occur 1 to 2 times per year (e.g. Cavicchia et al., 2014) and our results show that high-resolution oceanic models are necessary to represent the fine-scale features modulating the oceanic response. The atmospheric forcing should also be well resolved to realistically reproduce the intensity and variations of the surface input. The simple comparison with the oceanic impacts obtained by Renault et al. (2012) shows that Mediterranean storms are diverse and, depending on the area and season concerned, can lead to different oceanic responses due to the different processes. Previous studies at the climatological scale




show that the Sicily Strait is among the preferred areas for medicane formation (Cavicchia et al., 2014). The processes exposed here, especially the time lag between intense rainfall occurring at the beginning of the event and maximum intensity of heat and wind stress are characteristic of medicanes (Miglietta et al., 2013). We are thus convinced that the enhanced effect of the surface forcing and the reduced effect of the turbulent mixing due to the rain-induced barrier layer are rather widespread under medicanes. The ABV eddy, which significantly reduces the cooling due to colder SSTs and shallower MLD is not always

present. Yet, both cyclonic and anticyclonic structures are common in Central Mediterranean and able to modulate, as was shown here, the oceanic response to medicanes.

*Code and data availability.*   The source codes are available online: Meso-NH (http://mesonh.aero.obs-mip.fr/mesonh54), and NEMO (https://www.nemo-ocean.eu/); The PSY2V4R4 daily analyses are available on the Copernicus Marine Environment Monitoring Service portal (http://marine.copernicus.eu); The GHRSST satellite SST data can be obtained upon registration from the Centre de Recherche et d'Exploitation Satellitaire (CERSAT),

IFREMER (http://cersat.ifremer.fr)

*Author contributions.*   MNB and CLB designed the study and set up the numerical configuration. MNB performed the simulation. MNB and CLB analysed the results and wrote the paper.

*Competing interests.*   The authors declare no competing interests.

*Acknowledgements.*   This work is a contribution to the HyMeX program (Hydrological cycle in the Mediterranean EXperiment - http://www.hymex.org)

through INSU-MISTRALS support. The authors acknowledge the Pôle de Calcul et de Données Marines for the DATARMOR facilities (storage, data access, computational resources). The PSY2V4R4 daily analyses were made available by the Copernicus Marine Environment Monitoring Service (http://marine.copernicus.eu). The GHRSST satellite SST data were obtained from the Centre de Recherche et d'Exploitation Satellitaire (CERSAT), at IFREMER, Plouzane (France) on behalf of ESA/Medspiration project. The authors thank S. Jullien (LOPS) for valuable discussions.



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




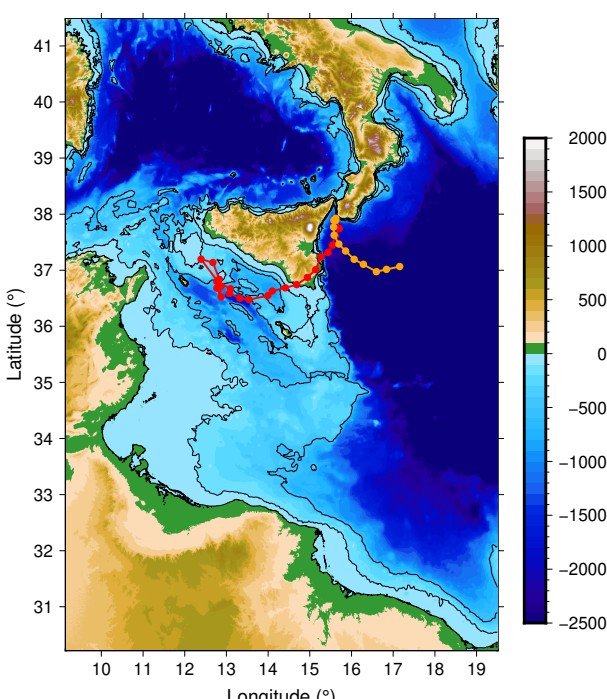

**Figure 1.** Map of the domain and bathymetry (m) used in the coupled simulation. The contours indicate the 100 m and 500 m isobaths. The track of the simulated medicane is indicated in red until 00 UTC on 8 November, in orange after that.



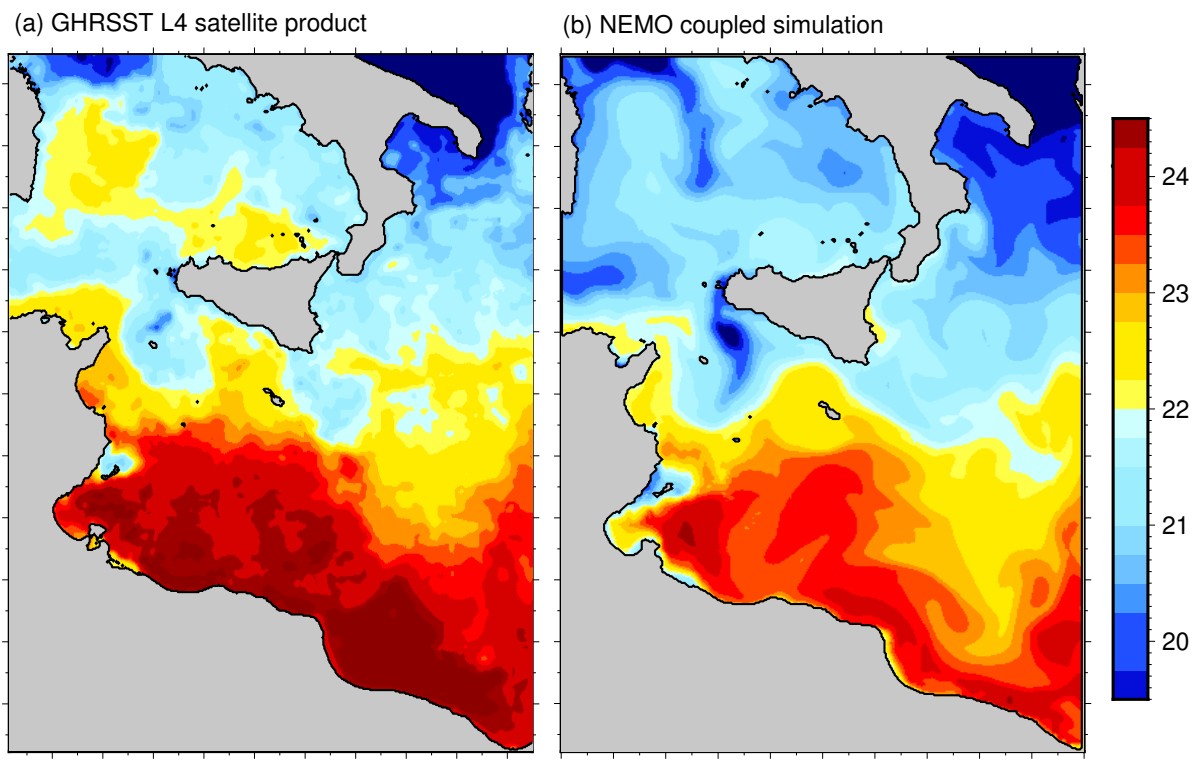

**Figure 2.** Map of the SST (°C) in the GHRSST L4 satellite product at 00 UTC (a) and in the output of the NEMO simulation at 01 UTC (b) on 7 November.



**Figure 3.** Map of the SI at 100 m (a), SSS (b), current at 15 m (c) and mixed layer depth (d), at the beginning of the simulation (04 UTC on 7 November). The black contours indicate the 100 and 500 m isobaths





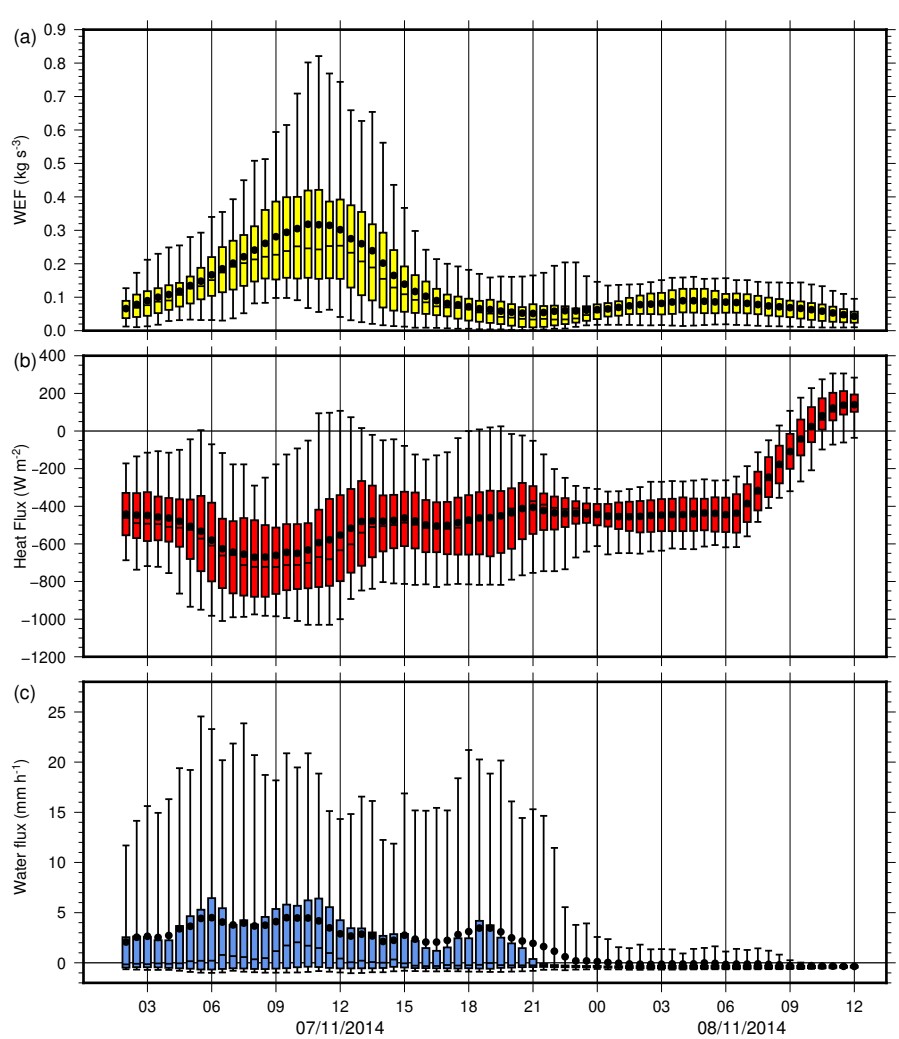

**Figure 4.** Time series of the median values of the wind energy flux (a), net heat flux (b) and water flux (positive values correspond to precipitation), (c) in the SFA. The boxes correspond to the 25 % / 75 % quantiles, the whiskers to the 5 % / 95 % quantiles. The solid black dots correspond to the mean values.


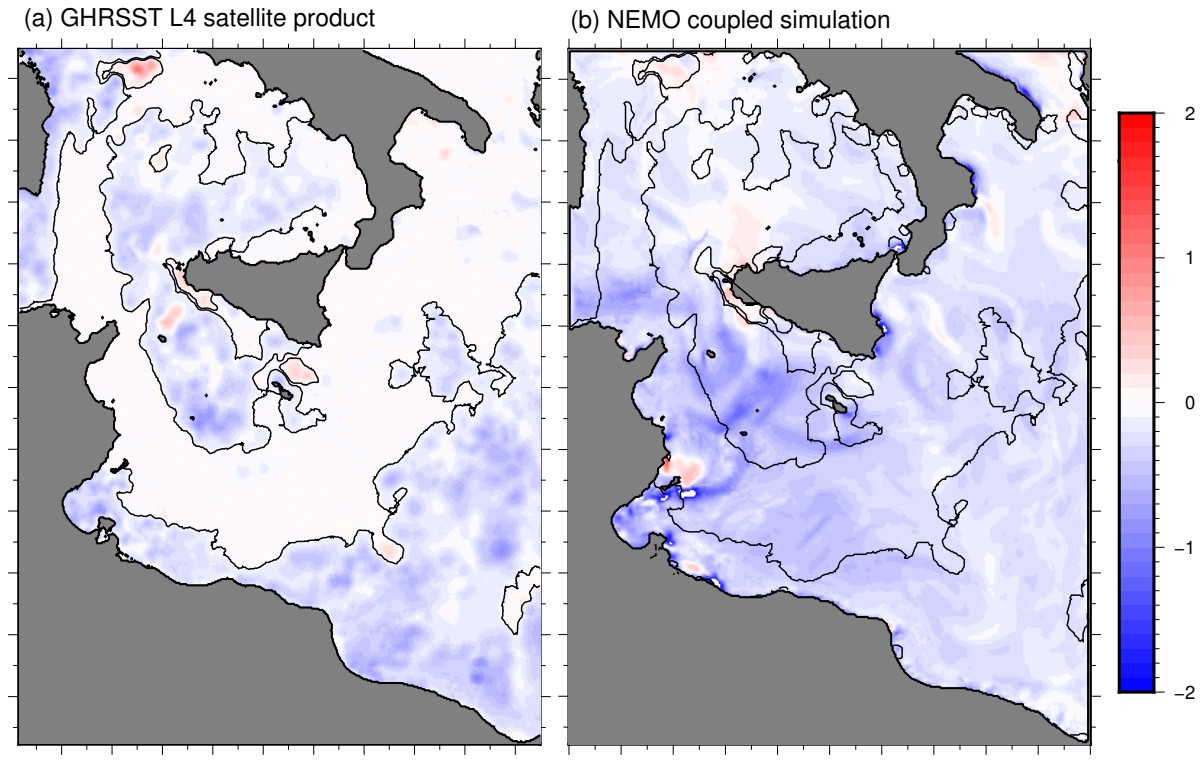

**Figure 5.** Map of the SST difference after 24 h (°C) in the GHRSST L4 satellite product (a) and in the output of the NEMO simulation (b). The white areas in (a) indicate the lack of satellite observations (reported also in (b) for a better comparison).

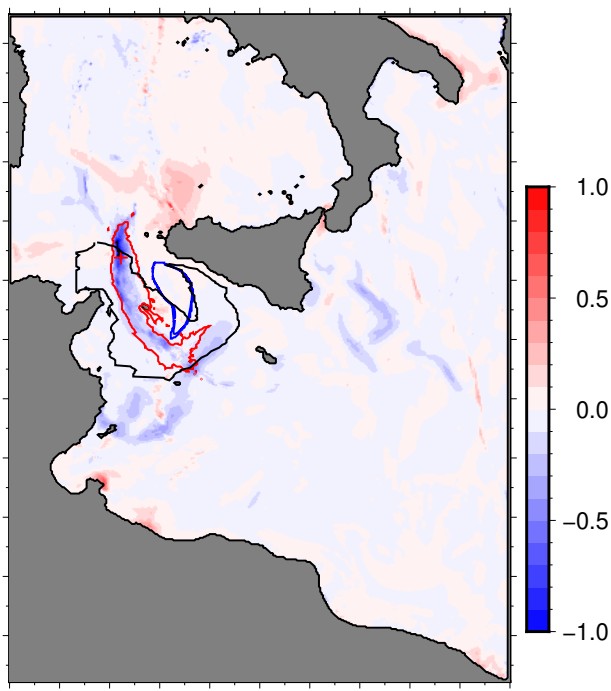

**Figure 6.** Map of the surface salinity difference at 00 UTC on 8 November, with respect to the beginning of the simulation. The black contours indicates the strong fluxes area, the blue contour the EDDY area and the red contour the HR area.





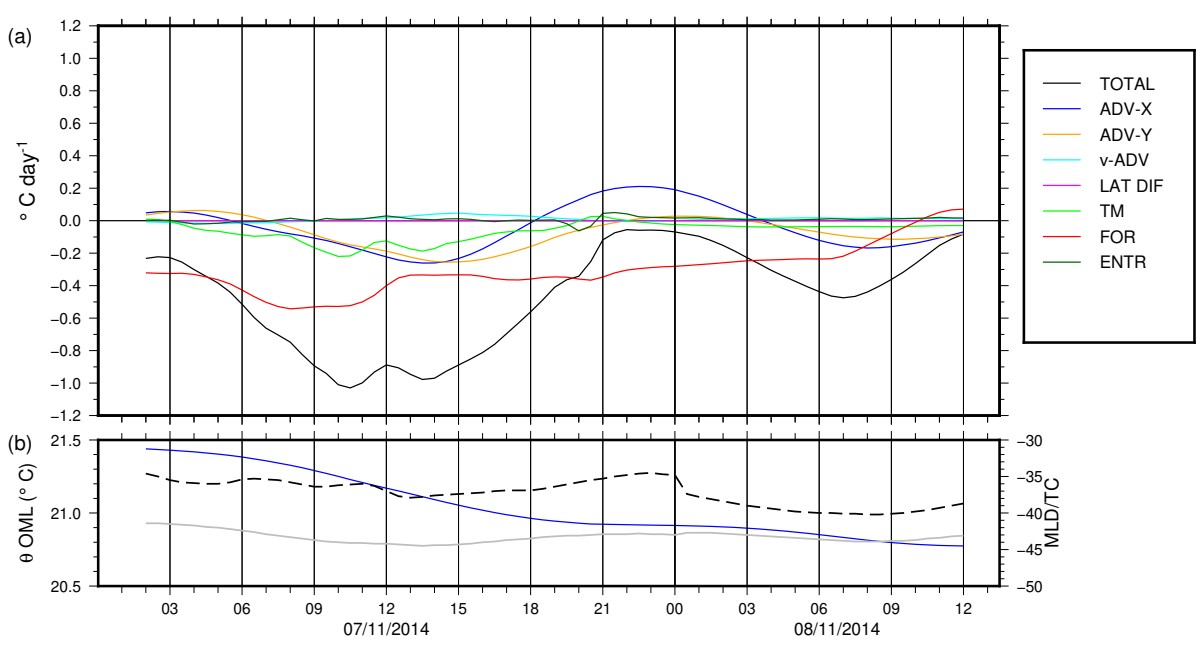

**Figure 7.** Time series of the different components of the tendency in the evolution of the potential temperature in the OML, in the SFA (a, colours), and time evolution of the temperature in the OML (blue), of the MLD (black dashed) and of the $T_{-10m} - 0.3°C$ thermocline (grey), in the SFA (b). The terms represented are horizontal advection (ADV-X and ADV-Y), the vertical advection v-ADV, the lateral diffusion LAT DIFF, the turbulent mixing TM, the surface heat forcing FOR and the entrainment at the bottom of the OML ENTR.



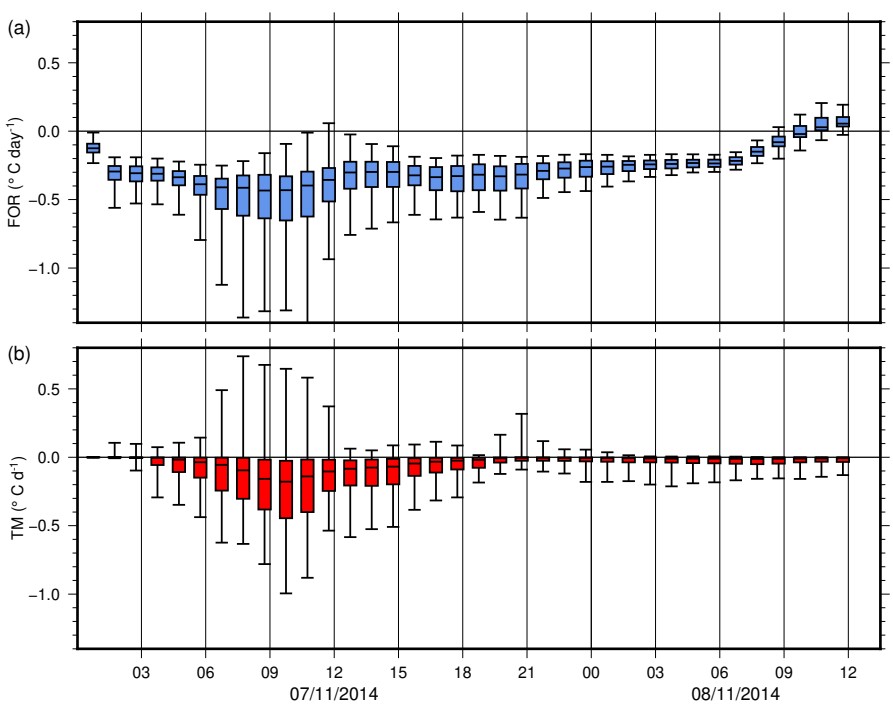

**Figure 8.** Time series of the median values of the surface forcing (FOR) (a) and turbulent mixing (TM) (b) tendency terms in the evolution of the temperature in the OML, in the SFA. The boxes correspond to the 25 % / 75 % quantiles, the whiskers to the 5 % / 95 % quantiles.





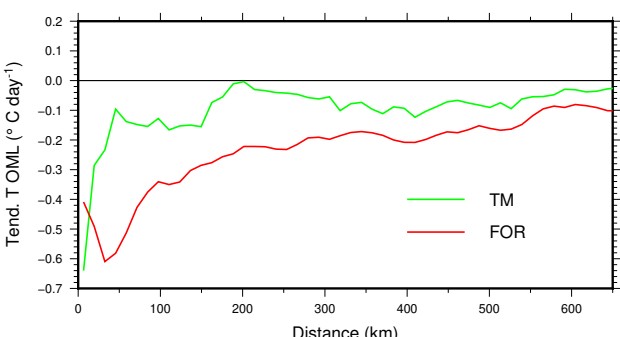

**Figure 9.** Mean values of the turbulent mixing (green) and surface forcing (red) tendency terms for temperature evolution in the OML at 10 UTC on 7 November. The values are radially averaged every 12 km from the medicane centre.

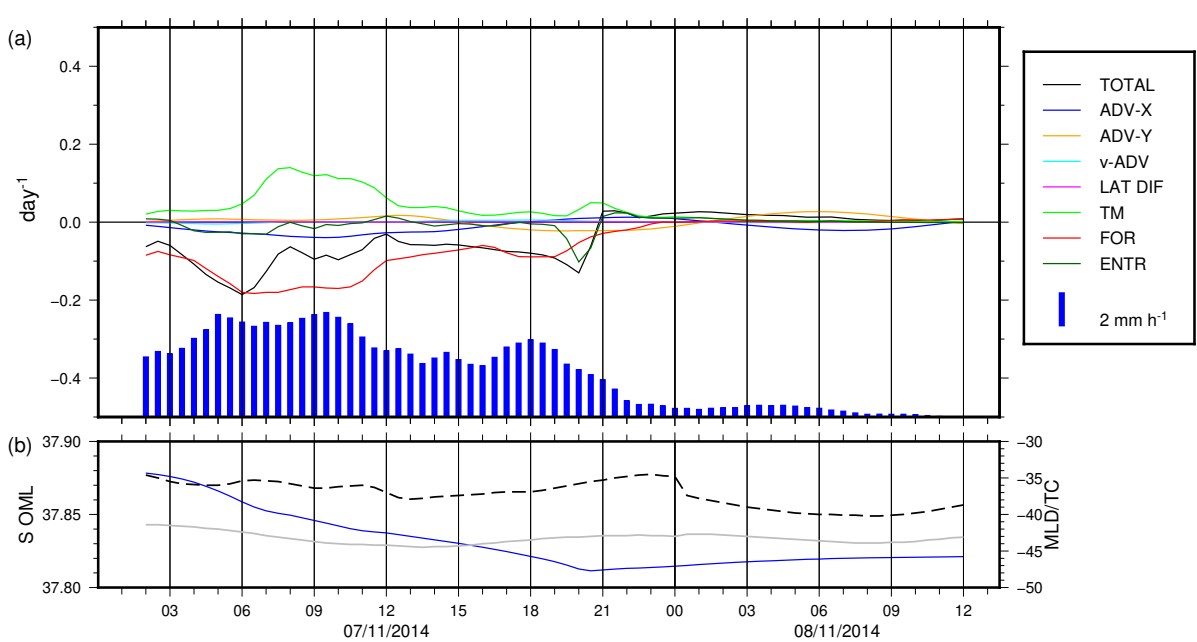

**Figure 10.** Time series of the different components of the tendency in the evolution of the salinity in the OML, in the SFA (a, colours) and of the instantaneous rain rate (blue), and time evolution of the salinity in the OML, of the MLD (black dashed) and of the $T_{-10m}-0.3°C$ thermocline (grey), in the SFA (b). The colours are as in Fig. 7.



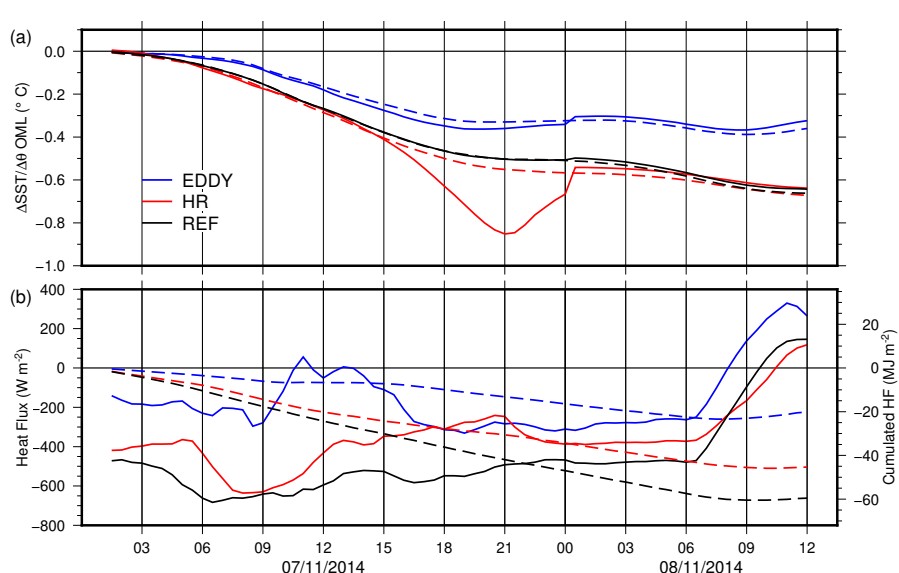

**Figure 11.** Time series of the SST (solid line) and potential temperature in the OML (dashed line) (a) and instantaneous (solid line) and integrated downward heat flux (dashed line) (b) in the EDDY (blue), HR (red) and REF (black) zones (see text)



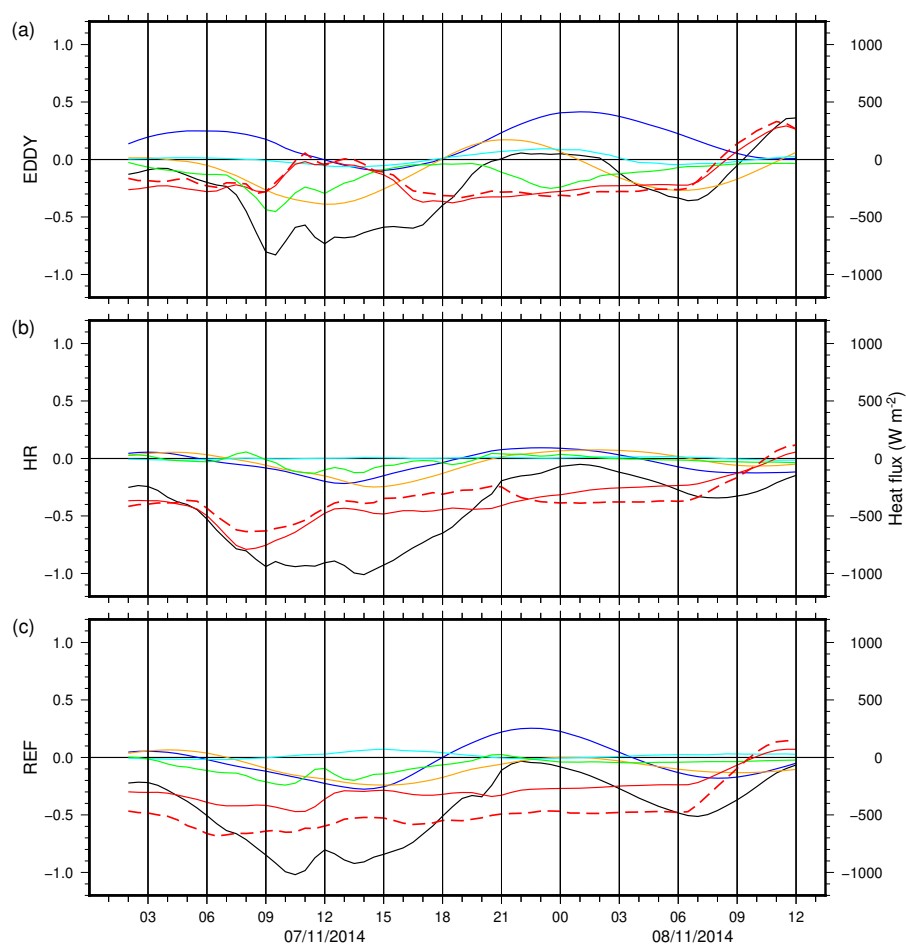

**Figure 12.** Time series of the different components of the tendency in the evolution of temperature in the OML, in the EDDY (a), HR (b), and REF areas (c). The colours are the same as in Fig. 7. Is also shown the net downward heat flux (dashed red).



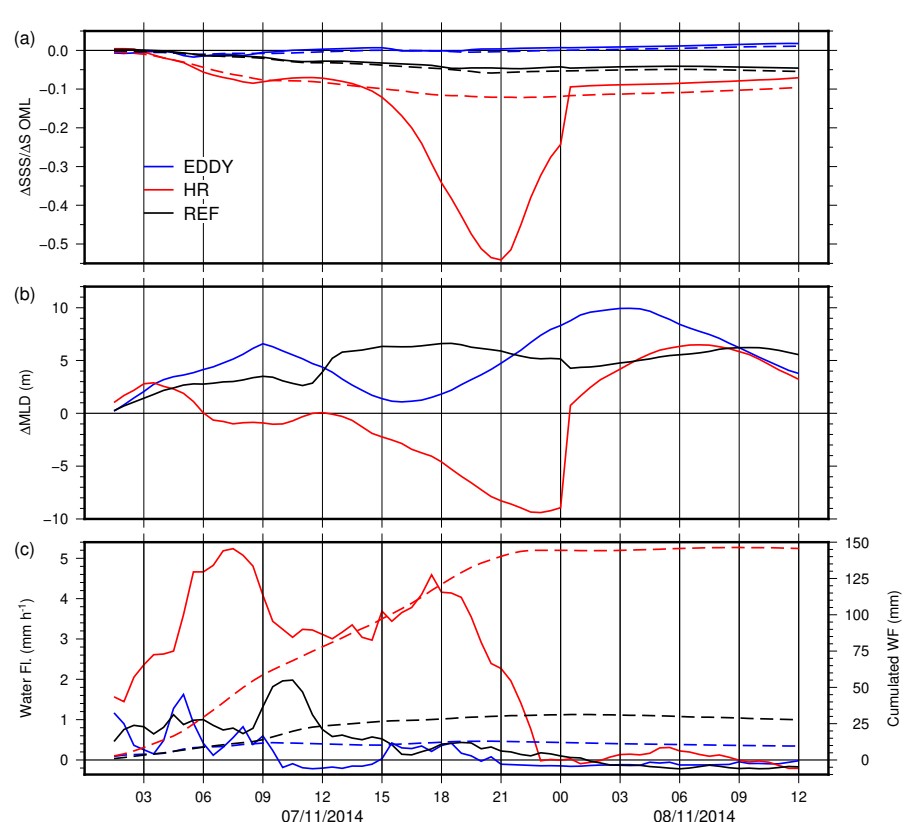

**Figure 13.** Time series of the SSS (solid line) and mean salinity in the OML (dashed line, a), MLD (b) and instantaneous (solid line) and integrated water flux (dashed line) (b) in the EDDY (blue), HR (red) and REF (black) zones (see text)



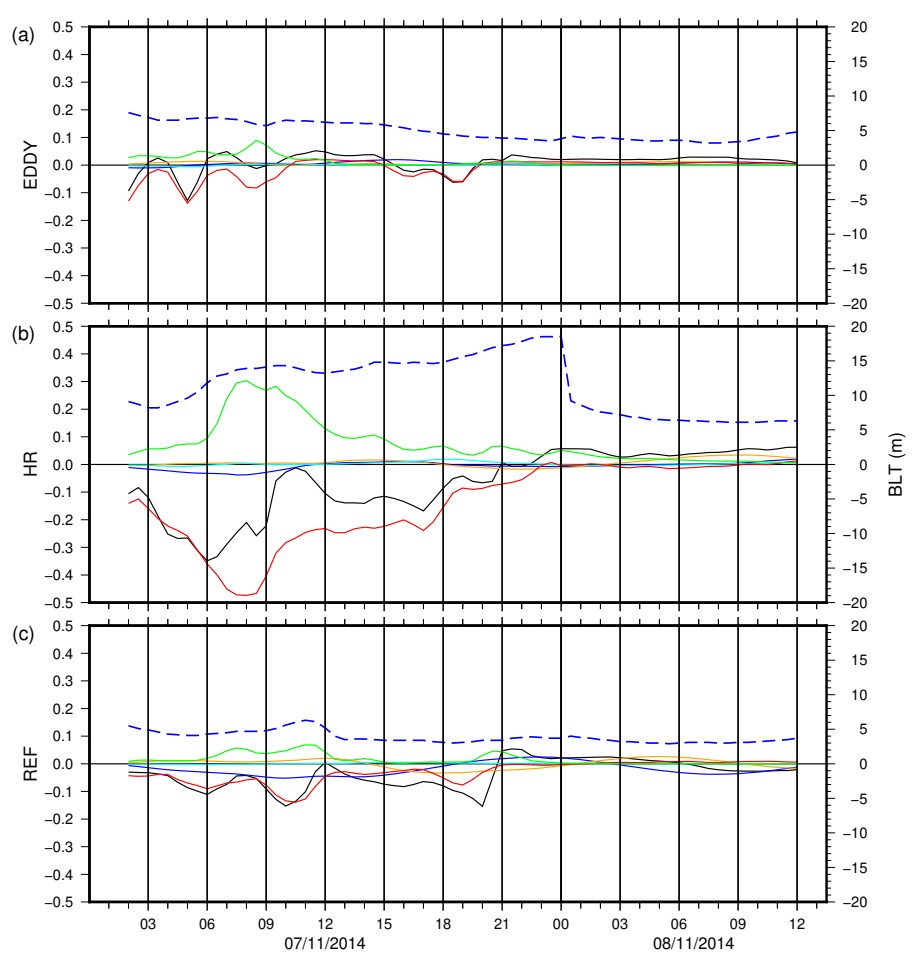

**Figure 14.** Time series of the different components of the tendency in the evolution of salinity in the OML, in the EDDY (a), HR (b), and reference areas (c). The colours are the same as in Fig. 7. Is also shown the barrier layer thickness (BLT) (dashed blue).





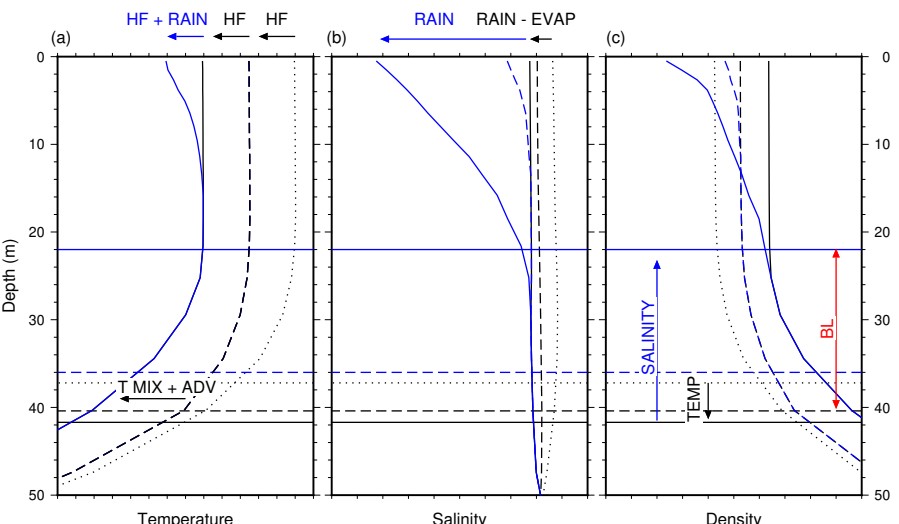

**Figure 15.** Diagram of the upper-layer evolution in temperature (a), salinity (b) and density (c) in the REF (black) and HR (blue, when it differs from REF) zones. The profiles of parameter and corresponding MLD are indicated at the beginning (dotted), middle (dashed) and end (solid) of the event.





**Table 1.** Oceanic parameters at the beginning of the simulation (04 UTC on 7 November) in the different zones (see text).

|  | SFA | EDDY | HR | REF |
|---|---|---|---|---|
| SST (°C) | $21.5 \pm 0.5$ | $20.0 \pm 0.4$ | $21.5 \pm 0.4$ | $21.6 \pm 0.5$ |
| $\langle \theta \rangle$ (°C) | $21.4 \pm 0.5$ | $19.9 \pm 0.4$ | $21.4 \pm 0.4$ | $21.5 \pm 0.5$ |
| SSS | $37.68 \pm 0.17$ | $37.82 \pm 0.05$ | $37.61 \pm 0.12$ | $37.73 \pm 0.17$ |
| $\langle S \rangle$ | $37.88 \pm 0.15$ | $38.01 \pm 0.04$ | $37.82 \pm 0.12$ | $37.91 \pm 0.17$ |
| MLD (m) | $33.8 \pm 11.3$ | $16.2 \pm 2.8$ | $28.9 \pm 9.1$ | $36.5 \pm 11.3$ |
| BLT (m) | $7.5 \pm 4.6$ | $8.0 \pm 2.3$ | $10.6 \pm 4.8$ | $6.1 \pm 3.9$ |
| SI (kg m$^{-2}$) | $125 \pm 14$ | $88 \pm 14$ | $126 \pm 16$ | $125 \pm 12$ |





**Table 2.** Evolution of oceanic parameters between the beginning of the simulation (04 UTC on 7 November) and 00 UTC on 8 November in the different zones (see text).

|  | SFA | EDDY | HR | REF |
| --- | --- | --- | --- | --- |
| $\Delta$SST (°C) | $-0.56 \pm 0.24$ | $-0.34 \pm 0.27$ | $-0.67 \pm 0.23$ | $-0.51 \pm 0.21$ |
| $\Delta\langle\theta\rangle$ (°C) | $-0.54 \pm 0.23$ | $-0.32 \pm 0.25$ | $-0.57 \pm 0.24$ | $-0.51 \pm 0.22$ |
| $\Delta$SSS | $-0.09 \pm 0.14$ | $0.01 \pm 0.03$ | $-0.24 \pm 0.22$ | $-0.04 \pm 0.07$ |
| $\Delta\langle S\rangle$ | $-0.07 \pm 0.07$ | $0.00 \pm 0.03$ | $-0.12 \pm 0.07$ | $-0.05 \pm 0.07$ |
| $\Delta$MLD (m) | $1.0 \pm 12.8$ | $8.3 \pm 4.0$ | $-8.9 \pm 13.7$ | $5.1 \pm 9.2$ |
| $\Delta$BLT (m) | $0.7 \pm 9.6$ | $-4.2 \pm 2.8$ | $7.8 \pm 12.7$ | $-2.4 \pm 4.7$ |

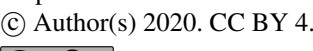



**Table 3.** Integrated forcings and effects of OML tendency terms between the beginning of the simulation (04 UTC on 7 November) and 12 UTC on 8 November in the different zones (see text).

|  | SFA | EDDY | HR | REF |
|---|---|---|---|---|
| HFL (MJ) | $-54.8 \pm 16.1$ | $-20.1 \pm 7.0$ | $-45.3 \pm 11.6$ | $-59.5 \pm 16.4$ |
| WFL (mm) | $63.0 \pm 61.9$ | $9.6 \pm 19.8$ | $136.0 \pm 36.4$ | $27.7 \pm 28.0$ |
| WEF ($10^3$kg s$^{-2}$) | $15.3 \pm 6.1$ | $4.7 \pm 3.6$ | $16.2 \pm 7.0$ | $14.3 \pm 6.2$ |
| INT FOR T (°C) | $-0.45 \pm 0.12$ | $-0.26 \pm 0.08$ | $-0.55 \pm 0.11$ | $-0.40 \pm 0.09$ |
| INT TM T (°C) | $-0.09 \pm 0.13$ | $-0.19 \pm 0.06$ | $-0.03 \pm 0.14$ | $-0.11 \pm 0.12$ |
| INT ADV-X T (°C) | $-0.06 \pm 0.22$ | $0.22 \pm 0.31$ | $-0.06 \pm 0.21$ | $-0.05 \pm 0.22$ |
| INT ADV-Y T (°C) | $-0.11 \pm 0.26$ | $-0.16 \pm 0.20$ | $-0.06 \pm 0.27$ | $-0.12 \pm 0.25$ |
| INT FOR S | $-0.09 \pm 0.10$ | $-0.02 \pm 0.04$ | $-0.21 \pm 0.10$ | $-0.04 \pm 0.05$ |
| INT TM S | $0.05 \pm 0.07$ | $0.02 \pm 0.03$ | $0.11 \pm 0.08$ | $0.02 \pm 0.04$ |
| INT ADV-X S | $-0.02 \pm 0.07$ | $0.01 \pm 0.02$ | $-0.01 \pm 0.07$ | $-0.03 \pm 0.08$ |
| INT ADV-Y S | $0.00 \pm 0.07$ | $0.01 \pm 0.03$ | $0.01 \pm 0.07$ | $0.00 \pm 0.07$ |