# Peer review of "Impact of a medicane on the oceanic surface layer from a coupled, kilometre-scale simulation"

_Ocean Science, 2020_

## Referee Comment (RC1) · James Hlywiak (Referee) · 29 May 2020

General Comments:

In this paper, the authors perform numerical simulations to show how Cyclone Qendreas (2014) impacted the upper ocean salinity and temperature structure near the Strait of Sicily. They show that the ocean response to wind stresses and the precipitation-driven water flux is similar to what occurs within weaker tropical cyclones, in that cooling within the oceanic mixed layer is mostly driven by radiative processes and less due to turbulent mixing below the thermocline. They also show how the ocean response varies in the vicinity of a cyclonic eddy and within a heavy rain region. The

work presented here is novel and highly impactful, and substantiates the existing body of literature regarding medicanes. Outside of a few minor grammar errors and typos, their interpretations of the results are mostly clear. My main critiques regard some inconsistencies in their analysis. Additionally, I would like an analysis about how well the model reproduces observations of the upper ocean. Therefore, I recommend minor revisions to be made based on my comments below.

Specific Comments:

- How well does the model simulate the initial upper ocean temperature and salinity profiles? You compare the NEMO SSTs with satellite observations in section 3.2, however nothing is said about the ability of the model to accurately reflect the vertical structure of this part of the Mediterranean.

- L236-240: add citations to back up what you say are typical flux values for TCs.

- Paragraph starting on L318, and shown in Fig. 9: Do you have an explanation for why the temperature tendency due to TM is stronger than FOR within 20 km? I'm surprised TM contributes so strongly at small radii. FOR is less surprising, as it seems to reach a maximum near the radius of maximum winds, where the wind stress is greatest.

- Following that last point, include what the RMW is in Fig. 9.

- L323: By "throughout the event", do you mean that the integrated effects of ADV-X,Y are to cool the OML by -0.12ËŽC?

- L329-L331 are also confusing and need clarification.

- L345: you say that the mean OML temperature increases between 04-07 UTC on the 7th, but looking back at Fig. 7, the mean temperature decreases throughout the simulation. Where do you see this increase?

- In section 4.2, you argue that the effect of precipitation on upper-ocean salinity drives the differences in cooling between HR and REF. However, from the previous paragraph

it seems that some significant stratification due to salinity is already present at the start of the simulation, as the difference in mean OML depth between HR and REF is roughly the mean width of the BL in SFA. Additionally, Fig. 3 suggests that much of the region is already highly stratified. Based on the analysis that follows in section 4, I don't doubt that precipitation from the cyclone further drives differences in heat fluxes between HR and REF. However, I interpret this as HR being already preconditioned to some extent at the start of the simulation, which made the OML in this region more sensitive to the precipitation-driven water flux, a feedback you acknowledge within the discussion. Would you agree with this interpretation?

- Similarly, you say in L394-395 that the surface salinity in HR is only "slightly" fresher by roughly 0.1 psu, however in your earlier analysis of Fig. 10 in section 3.3.2 you claim that the freshening of the OML by 0.1 psu between the 2-21UTC on the 7th is "significant" (L337). Therefore, if the latter freshening is significant, it seems that the initial difference in surface salinity would be significant, further indicating that the upper ocean within HR was preconditioned. Please clarify this discrepancy.

- Lastly, in regard to L85-87 in the introduction, Hlywiak and Nolan 2019 touches on the response of the barrier layer to TCs of category 1-2 strength (see the analysis of the weakest set of idealized, simulated TCs in a high shear, relatively low SST environment). This comment doesn't affect my thoughts expressed above, but it may be worthwhile to compare your results this paper.

Technical Corrections:

- L33: "Surface heat fluxes also act to cool the upper-ocean . . ."

- L47: "Colder SSTs . . . generate weaker cooling and limit the cyclone intensity". I suggest rephrasing this so that it doesn't sound like weaker cooling is the reason that cyclone intensity is limited.

- L58: Re-writing this line as ". . . the BL shoals the ML and isolates it from the colder

waters below, thus making surface heat extraction more efficient" or something similar would make it more grammatically correct

- L69-70 is a bit hard to read because there are too many commas

- L83: "we use of a coupled . . ."

- L93: "The present study aims to investigate. . ."

- L268: ". . . the sea surface height has decreased of 10 cm . . ."

- L301: ". . . in the SFA is given in Fig. 7 . . ."

- L327: ". . . forcing is higher than that of the turbulent mixing,. . ."

- L332: "contributes significantly . . ."

- L341: ". . . 7 November deepens . . ."

- L374: ". . . the WEF is much weaker in EDDY than in REF, as the integrated . . ."

- L439: ". . . deepens the OML by a few meters . . ."

- Fig 10: what is the actual rain rate? Include an additional axis or make this a new subplot.

- Figs 11, 12, 13, and 14 need legends

---

## Referee Comment (RC2) · Ali Harzallah (Referee) · 3 Jun 2020

Review of the paper

Impact of a medicane on the oceanic surface layer from a coupled, kilometre-scale simulation Marie-Noëlle Bouin and Cindy Lebeaupin Brossier

General comments

A high resolution model is used to study the impact of a medicane on the oceanic upper layer. The analyses of tendency terms show weaker cooling compared to tropical cyclones. Surface heat iňĆuxes dominate with upper-layer salinity decrease due to

heavy precipitation. The study also shows that the Strait dynamics may impact the role of advection in the medicane processes.

The paper addresses relevant scientific questions on medicane processes related to air-sea interactions in the highly dynamical area of the Strait of Sicily. Important conclusions are reached on the relative importance of the heat and mass fluxes at the sea surface and the vertical mixing in the water cooling of the upper layer.

Generally the text is well written. However, some typos remain to be corrected. Figures are well drawn and clear. If there are points to criticize would be the fact of using a single event and being based only on a numerical model. A quick validation of the model would have given more robustness to the conclusions even if they are largely convincing. The authors could say few words on the advantage of using a coupled model compared to a forced oceanic one. The authors could also present solid arguments on the ability of the model to reproduce the medicane, at least a comparison of the trajectories. A sentence could also be added regarding the added value of using a coupling frequency as high as 15 min.

I recommend minor corrections

Specific comments:

45. Could authors further precise how stronger stratification favour more intense cooling though turbulent mixing at the base of the OML.

126 The simulated medicane spent...Gulf of Gabès: This sentence needs reformulation to better illustrate the area bathymetry.

153. Could you please precise how error bounds are obtained and their significance in the presented comparison. This also may be the case for other values shown in the text.

160. Minimum values...: Please precise that values refer to MLD (SI values are not shown in the Gulf of Gabès).

242-255: Some change values are shown in Table 2, others not. This may be somewhat confusing.

251 : please precise that (-8.E10-9) is a salinity change.

252: According to Fig. 5b, salinisation is shown north of the Gulf of Gabès, not in the Gulf interior.

265: under the MLD : should be OML.

Figure captions: A further detailed description of the figures is recommended.

Fig.15 is a key figure in the paper. A more detailed description of this figure is recommended. Discussions of roles of heavy precipitation, preconditioning and cyclonic eddy may make reference to this figure.

Technical corrections

Some typing errors should be corrected

СЗ

---

## Short Comment (SC1) · 22 Jun 2020

This paper presents a study of the impact of a medicane on the oceanic upper layer using a high resolution model. The authors used a high coupling frequency of 15 min which is rather high. In our team, we worked on the impact of the wind forcing temporal resolution on the sea dynamics. Indeed, it is recommended to use a high temporal resolution. Could authors give few words on the added value of using such a high frequency (as high as 15 min)? If a lower frequency was used (for example hourly coupling) would the results be different?

Marwa Ouni PhD student Modelling team, INSTM

---

## Author Comment (AC2) · 10 Jul 2020

**OS-2020-38 - Response to reviewers**

We thank both reviewers for their constructive comments and rereading. We address them below.

**RC1 (J. Hlywiak)**

**Specific Comments:**

How well does the model simulate the initial upper ocean temperature and salinity profiles? You compare the NEMO SSTs with satellite observations in section 3.2, however nothing is said about the ability of the model to accurately reflect the vertical structure of this part of the Mediterranean.

This is a good remark, unfortunately, observations of temperature and salinity profiles in this part of the Mediterranean are scarce. Due to the strong dynamics and heavy maritime traffic in the zone, maintaining a surface long-term mooring is very demanding. From the Coriolis data website (http://www.coriolis.eu.org), only profiles several days before or after the event are available (see Fig. 1 below) in the Tyrrhenian Sea and Northern Ionian Sea. A single profile exists (at 10:36 UTC on 07 November - Fig. 2) and shows a good correspondence at first order with the model outputs and PSY2 analysis used as initial conditions. Indeed, to initialise the NEMO-SICIL36 model, we used the PSY2 global daily analysis that assimilated a large number of in situ and satellite data. As the vertical grid of the SICIL36 configuration is the same as PSY2, no vertical interpolation is done. The vertical structure of the ocean fields is thus mostly inherited from PSY2 and we think that there is good reliability o the analysed fields (despite maybe a too smooth thermocline, as already seen in Rainaud et al. 2017). But, as the ARGO profile is really out of the medicane track in the NW of the domain, we prefer not to use it here.

Rainaud, R., Lebeaupin Brossier, C., Ducrocq, V. and Giordani, H. (2017). High-resolution air-sea coupling impact on two heavy precipitation events in the Western Mediterranean. Q.J.R. Meteorol. Soc, 143, 2448-2462. doi:10.1002/qj.3098

Figure 1: Position of the ARGO floats with observations between 06 November 00UTC and 08 November 12UTC from the Coriolis data center http://www.coriolis.eu.org. Only the float 6901491 (light green) corresponds to the time of the simulation.

Figure 2: Position of the ARGO float 6901491 on 7 November, 10:36 UTC (yellow dot) and corresponding temperature and salinity profiles (from Coriolis, http://www.coriolis.eu.org, observations in black, NEMO model output in red, initial conditions from PSY2 Mercator analysis in dashed blue).

- L236-240: add citations to back up what you say are typical flux values for TCs. Done.

- Paragraph starting on L318, and shown in Fig. 9: Do you have an explanation for why the temperature tendency due to TM is stronger than FOR within 20 km? I'm surprised TM contributes so strongly at small radii. FOR is less surprising, as it seems to reach a maximum near the radius of maximum winds, where the wind stress is greatest.

First of all, this result should be interpreted with caution: only 2 points show a TM term larger than FOR because, as indicated in the legend of the figure, we used 12-km-radius circles here (corresponding to 5 grid points of the model). We looked for processes close to the bottom of the OML that could explain this cooling, like mixing, or an abrupt shoaling of the OML resulting in a change in the terms of the budget. There is no significant change of the MLD at that time and place, but the vertical velocities around 50 m (below the SST $-0.3^{\circ}$ C thermocline) are strong and show a radial profile close to that of the turbulent mixing term responsible for the cooling (Fig. 3 below). We added a comment about that effect in the text.

---

## Author Comment (AC1)

**OS-2020-38 - Response to short comment**
This paper presents a study of the impact of a medicane on the oceanic upper layer using a high resolution model. The authors used a high coupling frequency of 15 min which is rather high. In our team, we worked on the impact of the wind forcing temporal resolution on the sea dynamics. Indeed, it is recommended to use a high temporal resolution. Could authors give few words on the added value of using such a high frequency (as high as 15 min)? If a lower frequency was used (for example hourly coupling) would the results be different?
Marwa Ouni PhD student Modelling team, INSTM

Thank you for this comment. We did not test the impact of the coupling on the simulation before choosing a coupling frequency of 15 min. We did not test either the impact of using a fully coupled configuration with respect to an atmospheric model forcing the oceanic simulation (see our response to the referees' comments). From previous experiences on tropical cyclone, sea surge modelling, or other stormy events we inferred that the atmospheric forcing and its time evolution must be represented with a "sufficient" sampling. Sufficient here depends on the time step of the oceanic model (in our case 5 min), on its resolution ($1/36°$), on the scale of the processes we want to represent and on, typically, the translation speed of the event.
We think that quantifying the effect of using an hourly rather than a 15 min coupling is not directly feasible without testing the two configurations. However, first insights may be given by comparing the mean exchanged fields. For that, the coupled fields (wind stress components, total heat flux, net freshwater flux) can be inferred a posteriori by hourly averaging the 15 min exchanged fields. This permits to estimate the fields potentially transmitted to NEMO if a lower frequency had been chosen. In the present case, the Fig. 1 and 2 below show the time series of the mean values of the total heat flux on the strong flux area (SFA) and of the water flux on the heavy precipitation zone (HR) during the first 16 h of the event, sent by the atmospheric model to the oceanic model at 15 min or 1 h coupling frequency. It can be seen that, in addition to the smoothing effect due to averaging the field over 1 h versus 15 min, coupling at 1 h induces a time lag than can result in strong discrepancies when the field is evolving rapidly. For instance, at 11:30 UTC, the mean total heat flux extracted on the SFA is 92 W m$^{-2}$ stronger with a 1 h coupling than with a 15 min coupling. At 05:30 UTC, the mean water flux on HR is 2.4 mm h$^{-1}$ weaker with a 1 h coupling than with a 15 min coupling. Two snapshots corresponding to these examples (Fig. 3) show that local discrepancies are much larger than that.
The smoothing and lag effects were already highlighted by the sensitivity study to forcing frequency done by Lebeaupin Brossier et al. (2009) on a heavy precipitation event over the Mediterranean using the same atmospheric model but a different oceanic model. The main conclusion of this study is that even if the integrated exchanges at the interface are well represented in the forcing applied, the vertical diffusion could respond differently under severe and rapidly evolving conditions from that under moderate conditions, and consequently, the exchanges across the OML base could be different. In particular, a high coupling frequency (1 hour or less) appeared necessary to well reproduce the formation and the persistence of low salinity internal boundary layer which are very sensitive to the precipitation rate and the high precipitation duration. In summary, it cannot be concluded formally that using a coupling frequency of 1 h would significantly change the oceanic simulation, but these examples illustrate that the fluxes transmitted to the ocean are significantly changed in doing so and that the OML response may be underestimated, in particular under such short and intense meteorological event.

Lebeaupin Brossier, C., Ducrocq, V., and Giordani, H. (2009). Effects of the air–sea coupling time frequency on the ocean response during Mediterranean intense events. Ocean Dynamics, 59(4), 539-549.

[Figure]

Figure 1: Time series of the mean values of the total heat flux extracted from the ocean (W m$^{-2}$) in the SFA between 01 and 16 UTC on 7 November for the coupling frequencies of 15 min (black) and 1 h (red).

[Figure]

Figure 2: Time series of the mean values of the water flux transmitted to the ocean (mm h$^{-1}$) in HR between 01 and 16 UTC on 7 November for the coupling frequencies of 15 min (black) and 1 h (red).

[Figure]

Figure 3: Maps of the instantaneous differences of the total heat flux extracted from the ocean at 11:30 UTC (left) and of the water flux transmitted to the ocean at 05:30 UTC (right), between a 15 min coupling frequency and a 1 h coupling frequency (as estimated a posteriori). The black contours indicate the SFA and HR, respectively.

---

## Author Comment (AC3)

**OS-2020-38 - Response to reviewers**

We thank both reviewers for their constructive comments and rereading. We address them below.

**RC1 (J. Hlywiak)**

**Specific Comments:**

How well does the model simulate the initial upper ocean temperature and salinity profiles? You compare the NEMO SSTs with satellite observations in section 3.2, however nothing is said about the ability of the model to accurately reflect the vertical structure of this part of the Mediterranean.

This is a good remark, unfortunately, observations of temperature and salinity profiles in this part of the Mediterranean are scarce. Due to the strong dynamics and heavy maritime traffic in the zone, maintaining a surface long-term mooring is very demanding. From the Coriolis data website (http://www.coriolis.eu.org), only profiles several days before or after the event are available (see Fig. 1 below) in the Tyrrhenian Sea and Northern Ionian Sea. A single profile exists (at 10:36 UTC on 07 November - Fig. 2) and shows a good correspondence at first order with the model outputs and PSY2 analysis used as initial conditions. Indeed, to initialise the NEMO-SICIL36 model, we used the PSY2 global daily analysis that assimilated a large number of in situ and satellite data. As the vertical grid of the SICIL36 configuration is the same as PSY2, no vertical interpolation is done. The vertical structure of the ocean fields is thus mostly inherited from PSY2 and we think that there is good reliability o the analysed fields (despite maybe a too smooth thermocline, as already seen in Rainaud et al. 2017). But, as the ARGO profile is really out of the medicane track in the NW of the domain, we prefer not to use it here.

Rainaud, R., Lebeaupin Brossier, C., Ducrocq, V. and Giordani, H. (2017). High-resolution air–sea coupling impact on two heavy precipitation events in the Western Mediterranean. Q.J.R. Meteorol. Soc, 143, 2448-2462. doi:10.1002/qj.3098

[Figure]

Figure 1: Position of the ARGO floats with observations between 06 November 00UTC and 08 November 12UTC from the Coriolis data center http://www.coriolis.eu.org. Only the float 6901491 (light green) corresponds to the time of the simulation.

[Figure]

Figure 2: Position of the ARGO float 6901491 on 7 November, 10:36 UTC (yellow dot) and corresponding temperature and salinity profiles (from Coriolis, http://www.coriolis.eu.org, observations in black, NEMO model output in red, initial conditions from PSY2 Mercator analysis in dashed blue).

- L236-240: add citations to back up what you say are typical flux values for TCs.
Done.

- Paragraph starting on L318, and shown in Fig. 9: Do you have an explanation for why the temperature tendency due to TM is stronger than FOR within 20 km? I'm surprised TM contributes so strongly at small radii. FOR is less surprising, as it seems to reach a maximum near the radius of maximum winds, where the wind stress is greatest.

First of all, this result should be interpreted with caution: only 2 points show a TM term larger than FOR because, as indicated in the legend of the figure, we used 12-km-radius circles here (corresponding to 5 grid points of the model). We looked for processes close to the bottom of the OML that could explain this cooling, like mixing, or an abrupt shoaling of the OML resulting in a change in the terms of the budget. There is no significant change of the MLD at that time and place, but the vertical velocities around 50 m (below the SST$-0.3°$C thermocline) are strong and show a radial profile close to that of the turbulent mixing term responsible for the cooling (Fig. 3 below). We added a comment about that effect in the text.

[Figure]

Figure 3: Mean values of the turbulent mixing (green) and surface forcing (red) tendency terms for temperature evolution in the OML, and of the vertical velocity at 47 m depth (black, right-hand side axis) at 10 UTC on 7 November (mm s$^{-1}$). The values are radially averaged every 12 km from the medicane centre. The vertical dashed line indicates the radius of maximum wind. Note that the vertical velocity is represented positive downwards.

- Following that last point, include what the RMW is in Fig. 9.
Done

- L323: By "throughout the event", do you mean that the integrated effects of ADV-X,Y are to cool the OML by -0.12°C?
Yes. We clarified this.

- L329-L331 are also confusing and need clarification.
We reformulated "A case study in the Gulf of Mexico also showed that the surface heat fluxes induce a widespread, moderate cooling affecting the whole surface of the gulf, while vertical mixing results in a relatively stronger, more localized cooling around the cyclone centre (Morey et al., 2006)"

- L345: you say that the mean OML temperature increases between 04-07 UTC on the 7th, but looking back at Fig. 7, the mean temperature decreases throughout the simulation. Where do you see this increase?
Yes, this is a good remark. This sentence was referring to some points in the SFA where the OML temperature actually increases during this time - hence the reference to Fig. 8. But we feel that this is more confusing than relevant, so we removed that.

- In section 4.2, you argue that the effect of precipitation on upper-ocean salinity drives the differences in cooling between HR and REF. However, from the previous paragraph it seems that some significant stratification due to salinity is already present at the start of the simulation, as the difference in mean OML depth between HR and REF is roughly the mean width of the BL in SFA. Additionally, Fig. 3 suggests that much of the region is already highly stratified. Based on the analysis that follows in section 4, I don't doubt that precipitation from the cyclone further drives differences in heat fluxes between HR and REF. However, I interpret this as HR being already preconditioned to some extent at the start of the simulation, which made the OML in this region more sensitive to the precipitation-driven water flux, a feedback you acknowledge within the discussion. Would you agree with this interpretation?
Indeed, at the beginning of the simulation, the MLD is slightly larger in HR than in REF, and the few-meter difference corresponds roughly to the BLT difference. But the SI (at 100 m) is the same in HR and REF. Looking at the temperature/salinity/density anomaly profiles (Fig. 4 below) shows that the gradients in temperature and salinity in HR are more pronounced than in REF, that these differences roughly compensate each other and that the density profiles are almost similar in HR and REF. We added a comment on that in the text.

- Similarly, you say in L394-395 that the surface salinity in HR is only "slightly" fresher by roughly 0.1 psu, however in your earlier analysis of Fig. 10 in section 3.3.2 you claim that the freshening of the OML by 0.1 psu between the 2-21UTC on the 7th is "significant" (L337). Therefore, if the latter freshening is significant, it seems that the initial difference in surface salinity would be significant, further indicating that the upper ocean within HR was preconditioned. Please clarify this discrepancy.
Yes, this is a good remark. The surface salinity in HR is actually lower than in REF at the beginning of the simulation. The apparent discrepancy between the "significant" freshening and the "slight" gradient in the surface salinity comes from the fact that this area is marked by strong horizontal gradients. Thus, a salinity gradient of 0.1 or a temperature gradient of 1 °C have probably a rather weak impact on the dynamics. Oppositely, a time evolution of the same amount in a few hours (which is not due to horizontal advection, of course) is a marker of a strong response to the forcing. We replaced "significant freshening" by "slight freshening" in section 3.3.2 to

[Figure]

Figure 4: Mean vertical profiles of temperature, salinity and density anomaly in the EDDY (blue), HR (red) and REF (black) areas at the beginning of the simulation

make the text more homogeneous.

- Lastly, in regard to L85-87 in the introduction, Hlywiak and Nolan 2019 touches on the response of the barrier layer to TCs of category 1-2 strength (see the analysis of the weakest set of idealized, simulated TCs in a high shear, relatively low SST environment). This comment doesn't affect my thoughts expressed above, but it may be worthwhile to compare your results this paper.
Indeed, our results are consistent with the findings of this idealized study for weak TCs. We added a reference to that in the introduction - rather in the paragraph about the effect of the barrier layers than in the paragraph you suggested - and at the beginning of the discussion. Thank you for that.

**Technical Corrections:**

- L33: "Surface heat fluxes also act to cool the upper-ocean . . ."
Corrected.

- L47: "Colder SSTs . . . generate weaker cooling and limit the cyclone intensity". I suggest rephrasing this so that it doesn't sound like weaker cooling is the reason that cyclone intensity is limited.
Yes. We rephrased: "... result in weaker TCs and associated weaker cooling".

- L58: Re-writing this line as ". . . the BL shoals the ML and isolates it from the colder waters below, thus making surface heat extraction more efficient" or something similar would make it more grammatically correct
Done, thank you.

- L69-70 is a bit hard to read because there are too many commas - L83: "we use of a coupled ..."
We cut the sentence in two and rephrased it.

- L93: "The present study aims to investigate. . ."
Corrected.

- L268: ". . . the sea surface height has decreased by 10 cm . . ."
Corrected.

- L301: "... in the SFA is given in Fig. 7 ..."
Corrected.

- L327: ". . . forcing is higher than that of the turbulent mixing,. . ."
Corrected.

- L332: "contributes significantly . . ."
Corrected.

- L341: ". . . 7 November deepens . . ."
Corrected.

- L374: ". . . the WEF is much weaker in EDDY than in REF, as the integrated . . ."
Corrected.

- L439: "... deepens the OML by a few meters ..."
Corrected.

- Fig 10: what is the actual rain rate? Include an additional axis or make this a new subplot.

We included a subplot.

- Figs 11, 12, 13, and 14 need legends
Done.

**RC2 (A. Harzallah)**

**General Comments:**

If there are points to criticize would be the fact of using a single event and being based only on a numerical model. A quick validation of the model would have given more robustness to the conclusions even if they are largely convincing.
We agree that a validation of the in-depth behavior of the model would add value to our results. Unfortunately, no observation profiles are available close to the medicane track for the period of the simulation (see response to RC1).

The authors could say few words on the advantage of using a coupled model compared to a forced oceanic one.
This is a good remark. We are not convinced that the same configuration, with the same exchange frequency but an oceanic model forced by the atmospheric model would lead to significantly different results. A previous study using the very same configuration on the same case study, on the atmospheric modelling of the event showed that the SST (and surface currents) feedback on the atmosphere is very weak (Bouin and Lebeaupin Brossier, 2020). So, the full coupling versus forcing probably does not change the oceanic evolution in this case. We used the fully-coupled configuration because it was originally developed to investigate the impact of the ocean on the atmosphere. We added a comment on that in the text (section 2.2.3).
Bouin, M. N., and Lebeaupin Brossier, C. (2020). Surface processes in the 7 November 2014 medicane from air-sea coupled high-resolution numerical modelling. Atmospheric Chemistry and Physics, 20, 6861-6881.

The authors could also present solid arguments on the ability of the model to reproduce the medicane, at least a comparison of the trajectories.
The validation of the atmospheric part of the simulation, including the comparison of the medicane track with observations is done in Bouin and Lebeaupin Brossier (2020). We added a reference to that in the text.

A sentence could also be added regarding the added value of using a coupling frequency as high as 15 min.
Yes, we added a sentence on the choice of the coupling frequency - see also the response to the interactive comment by Marwa Ouni.

**Specific Comments:**

- 45. Could authors further precise how stronger stratification favour more intense cooling though turbulent mixing at the base of the OML.
Stronger stratification corresponds to a larger gradient between the SST and the temperature below the thermocline. If the storm is sufficiently intense, the mixing due to wind stress can break this stratification and bring this - much colder - water at the surface, resulting in a stronger SST cooling. We added a reference in the text.

- 126 The simulated medicane spent...Gulf of Gabès: This sentence needs reformulation to better illustrate the area bathymetry.
Yes, we reformulated and it reads: "Also, large coastal areas are shallower than 100 m, for instance in the Gulf of Gabès. This shallow bathymetry makes the area very sensitive to surface forcing and lateral advection".

- 153. Could you please precise how error bounds are obtained and their significance in the presented comparison. This also may be the case for other values shown in the text.
They correspond to one sigma (standard deviation) values and this is now specified.

- 160. Minimum values...: Please precise that values refer to MLD (SI values are not shown in the Gulf of Gabès).
Done.

- 242-255: Some change values are shown in Table 2, others not. This may be somewhat confusing.
Only the mean values of the evolution of the parameters are given in Table 2, but we felt that given additional values (minimum, maximum) in text illustrates the variability of the response of the ocean. We tried to clarify this.

- 251 : please precise that (-8.E10-9) is a salinity change.
Done.

- 252: According to Fig. 5b, salinisation is shown north of the Gulf of Gabès, not in the Gulf interior.
Yes, this has been corrected.

- 265: under the MLD : should be OML.
Yes, corrected.

- Figure captions: A further detailed description of the figures is recommended.
We added details in the figure captions and figure legends when missing.

- Fig.15 is a key figure in the paper. A more detailed description of this figure is recommended. Discussions of roles of heavy precipitation, preconditioning and cyclonic eddy may make reference to this figure.

We discussed the results in more details using this figure (in the discussion), we also completed the legend of this figure. Thank you for that.

**Technical Corrections:**

Some typing errors should be corrected

We proofread and corrected the text.

---

## Referee Report (RR1)

**Second review of the paper**

**Impact of a medicane on the oceanic surface layer from a coupled, kilometre-scale simulation**

Marie-Noëlle Bouin and Cindy Lebeaupin Brossier

This version has been improved compared to the previous one. The authors have taken into account the comments made during the first review.

One minor point;

Line 250:  please verify   "and SI"   in relation with the variables shown in Table 2.

No need for another review.

I recommend **accept**